# Targeted Reduction of Causal Models

**Armin Kekić**[1]                **Bernhard Schölkopf**[1]                **Michel Besserve**[1]

[1]Max Planck Institute for Intelligent Systems, Tübingen, Germany

## Abstract

Why does a phenomenon occur? Addressing this question is central to most scientific inquiries and often relies on simulations of scientific models. As models become more intricate, deciphering the causes behind phenomena in high-dimensional spaces of interconnected variables becomes increasingly challenging. Causal Representation Learning (CRL) offers a promising avenue to uncover interpretable causal patterns within these simulations through an interventional lens. However, developing general CRL frameworks suitable for practical applications remains an open challenge. We introduce *Targeted Causal Reduction* (TCR), a method for condensing complex intervenable models into a concise set of causal factors that explain a specific target phenomenon. We propose an information theoretic objective to learn TCR from interventional data of simulations, establish identifiability for continuous variables under shift interventions and present a practical algorithm for learning TCRs. Its ability to generate interpretable high-level explanations from complex models is demonstrated on toy and mechanical systems, illustrating its potential to assist scientists in the study of complex phenomena in a broad range of disciplines.[1]

## 1 INTRODUCTION

Numerical models are indispensable in science for simulating real-world systems and generating *etiological explanations*—identifying the causes of specific phenomena. General circulation models, for example, shed light on the causes of global warming (Grassl, 2000), while com-

---

[1]Code is available at: https://github.com/akekic/targeted-causal-reduction.git.

putational brain models explore the origins of neurological pathologies (Breakspear, 2017; Deco and Kringelbach, 2014). These examples illustrate the increasing complexity of numerical scientific models, designed to faithfully capture the large number of mechanisms at play in these systems. However, this complexity comes at a cost: expanding parameter spaces and heightened computational demands. This trend, in turn, impacts the ability to generate high-level explanations, understandable by scientists and decision makers (Reichstein et al., 2019; Safavi et al., 2023).

Effective human explanations are often based on understanding a few causal relations between a limited number of variables. While the simulation of complex systems might rely on numerous simple mechanisms, extracting overarching causal relations between fewer relevant high-level variables remains largely unaddressed. In particular, while causal representation learning tries to explain data based on a learned latent causal graph (Wendong et al., 2023; Squires et al., 2023; von Kügelgen et al., 2023a), it currently has theoretical and practical limitations. CRL largely relies on preserving all information in the data to provide recoverability guarantees for the latent causes, while the idea of a high-level representation is precisely to discard irrelevant data.

In contrast, Causal Model Reduction (CMR), which aims to map a low-level causal model to a simpler high-level model with fewer or lower-dimensional variables, embraces the purpose of eliminating irrelevant information. However, existing CMR approaches, such as causal abstractions (Geiger et al., 2023; Zennaro et al., 2023) and Causal Feature Learning (CFL) (Chalupka et al., 2016) are not well-suited to causally describe many scientific models: they use discrete variables and typically rely on *hard* interventions, disconnecting causal variables from their parents. The following example shows, however, that simpler high-level causal models for continuous variables and *soft* interventions are natural and useful in domains such as physics.

Consider a system of point masses connected by springs shown in Fig. 1a, where each mass is influenced by random

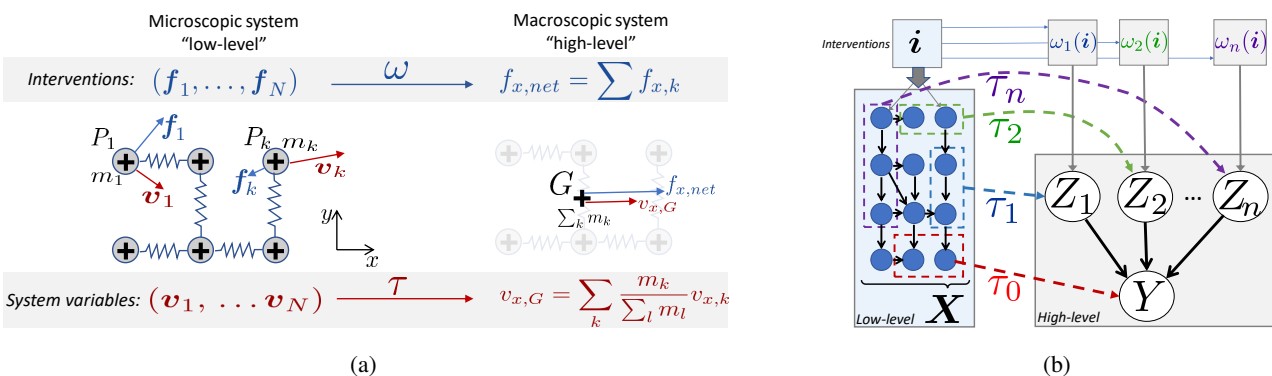

Figure 1: **Targeted Causal Reduction.** (a) Example targeted model reduction: a model of the dynamics of a system of point masses connected by springs can be reduced to the trajectory of its center of mass. (b) Overview of TCR. Low-level variables $X$ (simulation) are mapped to high-level variables $(Z, Y)$ with a fixed causal structure. The target $Y$ is known, while the causes $Z$ and the high-level causal mechanism are learned. Additionally, we learn a mapping from low-level shift interventions $i$ to high-level shift interventions $\omega(i)$.

external forces. Its trajectory under the intervention of external forces can be accurately predicted by simulating the coupled equations of motion of individual point masses. However, if we are only interested in a particular macroscopic "target" variable of this system: the horizontal speed of the system's center of mass at the end of an experiment, a key result from classical physics is that its motion will depend only on the sum of all horizontal components of external forces applied over time. We thus obtain a form of CMR: a much simpler system that accurately accounts for the effect of interventions in the system on the target variable.

This highlights the core elements needed for CMR in scientific models: (1) there is a clearly defined macroscopic target variable, (2) continuous low-level variables are reduced to a smaller set of continuous high-level variables, and (3) low-level interventions are soft: exerted forces modify the future trajectory but do not suppress the influence of other factors such as the past state of the system. Moreover, it is the combination of many low-level soft interventions across time that corresponds to a relevant high-level intervention. These three aspects are commonly found characteristics of studied real-world systems, motivating the development of CMR algorithms adapted to this setting.

In this paper, we introduce *Targeted Causal Reduction* (TCR), depicted in Fig. 1b, a novel approach designed to simplify complex *low-level models* into *high-level models*, focused on explaining causal influences on an observable target variable $Y$. The key signal we use for learning are *interventions* applied to the low-level variables, which are mapped to high-level interventions in a way that captures the causal influences on $Y$ in a concise and interpretable way. We formulate this learning objective as a Kullback-Leibler divergence between the fitted high-level interventional model and the reduced low-level interventional distribution, leading to a practical learning algorithm for the case of linear reductions. Applications to high-dimensional synthetic and scientific models demonstrates accuracy and

interpretability of our approach. We refer to the appendix for more related work (A) and proofs (C).

## 2 BACKGROUND

### 2.1 STRUCTURAL CAUSAL MODELS

Causal dependencies between variables can be described using *Structural Causal Models* (SCM) (Peters et al., 2017).

**Notation.** We use boldface for column vectors, and $i_S$ for the subvector of $i$ restricted to the components in set $S$.

**Definition 2.1** (SCM). *An n-dimensional structural causal model is a triplet $\mathcal{M} = (\mathcal{G}, \mathbb{S}, P_U)$ consisting of:*

- *a joint distribution $P_U$ over exogenous variables $\{U_j\}_{j \leq n}$,*
- *a directed graph $\mathcal{G}$ with n vertices,*
- *a set $\mathbb{S} = \{X_j := f_j(\boldsymbol{Pa}_j, U_j), j = 1, \ldots, n\}$ of structural equations, where $\boldsymbol{Pa}_j$ are the variables indexed by the set of parents of vertex j in $\mathcal{G}$,*

*such that for almost every $\boldsymbol{u}$, the system $\{x_j := f_j(\boldsymbol{pa}_j, u_j)\}$ has a unique solution $\boldsymbol{x} = \boldsymbol{g}(\boldsymbol{u})$, with $\boldsymbol{g}$ measurable.*

The unique solvability condition is included in this definition because we consider a general class of SCMs by allowing cycles, that is, $\mathcal{G}$ may not be a DAG. Moreover, we allow hidden confounding through the potential lack of independence between the exogenous variables $\{U_j\}$. See Bongers et al. (2021) for a thorough study of these models. Under these conditions, the distribution $P_U$ entails a well-defined joint distribution over the endogenous variables $P(X)$.

Interventions in SCMs involve replacing one or more structural equations, potentially modifying exogenous distributions, and adding or removing arrows in the original graph to reflect changes in dependencies between variables. An

intervention transforms the original model $\mathcal{M} = (\mathcal{G}, \mathbb{S}, P_U)$ into an intervened model $\mathcal{M}^{(i)} = (\mathcal{G}^{(i)}, \mathbb{S}^{(i)}, P_U^{(i)})$, where $i$ is the vector parameterizing the intervention. The base probability distribution of the unintervened model is denoted $P_{\mathcal{M}}^{(0)}(X)$ or simply $P_{\mathcal{M}}(X)$ and the interventional distribution associated with $\mathcal{M}^{(i)}$ is denoted $P_{\mathcal{M}}^{(i)}(X)$.

Classical *do*-interventions set a structural equation to a constant, removing all influences of the parents on the intervened variable. This can be problematic for studying how the influence of low-level variables is propagated to the target, since for simultaneous interventions, the effects of some interventions can be masked by others. The probability of such masking increases as the number of low-level variables grows. *Soft interventions*, on the other hand, modify an equation while keeping the set of parents unchanged. This is more appropriate in our setting, since it propagates the information from all interventions to the target simultaneously.

Large classes of soft interventions can be designed to match domain knowledge (Besserve and Schölkopf, 2022). Notably, *shift interventions* modify the structural equation of endogenous variable $l$ through shifting it by a scalar parameter $i$

$$\{X_l := f_l(\mathbf{Pa}_l, U_l)\} \mapsto \{X_l := f_l(\mathbf{Pa}_l, U_l) + i\}. \quad (1)$$

These can be combined to form multi-node interventions with vector parameter $i$.

## 2.2 SIMULATIONS AND CAUSAL MODELS

We use the term *scientific model* to refer to a generative model that relies on a set of equations to represent a phenomenon. What distinguishes such models from generative models in machine learning is their decomposability into elementary functions, encoding domain knowledge about the mechanisms being investigated. Simulations based on the numerical solution of scientific models can often be expressed as SCMs; This notably includes Ordinary (ODE) (Mooij et al., 2013) and Stochastic Differential Equations (SDE) (Hansen and Sokol, 2014). A simulator can thus be seen as a low-level causal model, from which samples of unintervened and intervened distributions can be generated. This forms the basis of the causal framework for learning high-level explanations for simulators developed in Sec. 3.1.

## 2.3 CAUSAL MODEL REDUCTIONS (CMR)

We consider as CMR any (possibly approximate) mapping from a low-level SCM $\mathcal{L}$ to a simpler high-level SCM $\mathcal{H}$. An example is CFL (Chalupka et al., 2015, 2016), which achieves a CMR by merging values of a large observation space to yield discrete high-level variables taking values in a small finite set. Consider:

- $\mathcal{L}$ has a vector of endogenous variables $X$ with range $\mathcal{X}$ and a set of interventions $\mathcal{I}$,
- $\mathcal{H}$ has a vector of endogenous variables $Z$ with range $\mathcal{Z}$ and a set of interventions $\mathcal{J}$.

Starting from the distribution of the low-level model $P_{\mathcal{L}}(X)$, a deterministic mapping $\tau : \mathcal{X} \rightarrow \mathcal{Z}$ generates a joint distribution on the high-level variables that is the push-forward distribution of $P_{\mathcal{L}}(X)$ by $\tau$, denoted $\tau_{\#}[P_{\mathcal{L}}(X)]$ such that

$$\tau(X) \sim \tau_{\#}[P_{\mathcal{L}}(X)].$$

The low-level interventional distributions can be pushed forward to the high-level in the same way.

A general framework for CMR is based on the notion of *exact transformation*, which ensures *interventional consistency* by matching the push-forward low-level distributions to the high-level ones.

**Definition 2.2** (Exact transformation (Rubenstein et al., 2017)). *A map $\tau : \mathcal{X} \rightarrow \mathcal{Z}$ is an exact transformation from $\mathcal{L}$ to $\mathcal{H}$ if it is surjective, and there exists a surjective intervention map $\omega : \mathcal{I} \rightarrow \mathcal{J}$ such that for all $i \in \mathcal{I}$*

$$\tau_{\#}[P_{\mathcal{L}}^{(i)}(X)] = P_{\mathcal{H}}^{(\omega(i))}(Z).$$

The set of possible $\tau$ can be restricted to *constructive transformations*, where high-level variables depend only on non-overlapping subsets of low-level variables. This eases interpretability of CMR and comes with characterization results (Beckers and Halpern, 2019; Geiger et al., 2023).

**Definition 2.3.** *$\tau : \mathcal{X} \rightarrow \mathcal{Z}$ is a constructive transformation between model $\mathcal{L}$ and $\mathcal{H}$ if there exists an alignment map $\pi$ relating indices of each high-level endogenous variable to a subset of indices of low-level endogenous variables such that for all $k \neq l$, $\pi(k) \cap \pi(l) = \emptyset$ and for each component $\tau_k$ of $\tau$ it exists a function $\bar{\tau}_k$ such that for all $x$ in $\mathcal{X}$,*

$$\tau_k(x) = \bar{\tau}_k(x_{\pi(k)}).$$

The intervention map $\omega$ of constructive exact transformations are required to be constructive as well, such that acting on high-level variable $k$ depends only on low-level interventions acting on variables in $\pi(k)$ (see App. B).

## 3 THEORETICAL ANALYSIS

As described in Fig. 1b, we consider endogenous variables of a low-level model gathered in a (high-dimensional) random vector $X$. A target scalar variable $Y = \tau_0(X)$ quantifies a property of interest of this model, and can be thought of as quantifying the presence or magnitude of a *phenomenon* in the data, using *detector* $\tau_0$. To generate a high-level causal explanation of this phenomenon, we learn a high-level SCM with a fixed causal structure, where the known effect variable $Y$ is caused by $n$ learned independent high-level

variables $Z_k$. The low-level variables $X$ are approximately mapped to the high-level variables $Z$ using a constructive transformation and an associated constructive interventional map with the same alignment $\pi$.

## 3.1 TCR FRAMEWORK

Our reduction framework has the following elements:

(1) A low-level SCM $\mathcal{L}$ with $N$ endogenous variables $\{X_1, \ldots, X_N\}$ and corresponding exogenous variables $\{U_k\}_{k=1..N}$ equipped with joint distribution $P(U)$. A set of low-level shift interventions parameterized by vector $i \in \mathcal{I}$ with distribution $P(i)$, with each component $i_k$ acting on one endogenous variable $X_k$. We only assume we can sample from unintervened and interventional distributions of $\mathcal{L}$.

(2) A class of high-level SCMs $\{\mathcal{H}_\gamma\}_{\gamma \in \Gamma}$ with (n+1) endogenous variables $\{Y, Z_1, \ldots, Z_n\}$ and associated exogenous variables $\{R_k\}_{k=0..n}$, equipped with a factorized distribution $P(R) = \prod P_{R_k}$. A set of high-level shift interventions parametrized by vector $j \in \mathcal{J}$, with each component $j_k$ affecting a single node $Z_k$. In contrast to the (fixed) low-level model, the high-level model parameters $\gamma$ are learned.

These two levels are linked by a constructive transformation with two deterministic surjective maps $\tau$ and $\omega$ from low- to high-level endogenous variables and interventions, respectively, which decompose as

$$\tau = (\tau_0, \tau_1, \tau_2, \ldots, \tau_n) \text{ with } \tau_k : x \mapsto \bar{\tau}_k(x_{\pi(k)}) \quad (2)$$

$$\omega = (\omega_0, \omega_1, \omega_2, \ldots, \omega_n) \text{ with } \omega_k : i \mapsto \bar{\omega}_k(i_{\pi(k)}) \quad (3)$$

where $\pi$ is a so-called alignment function from $[0..n]$ to non-overlapping subsets of $[1..N]$. Importantly, $\tau_0$ (and thus $(\bar{\tau}_0, \pi(0))$) are assumed fixed and known. Additionally, $\omega_0$ is assumed to be a trivial constant map $i \to 0$, to ensure that the high-level target variable cannot be directly intervened upon, as we want to explain the changes in $Y$ exclusively through changes of its high-level causes.

A high-level model involves the following mechanisms, which need to be learned: (1) The marginal distribution of each high-level cause $P^{(j)}(Z_k)$ in all high-level interventional settings $j$. (2) The mechanism $P(Y|Z)$ mapping high-level causes to $Y$, comprised of the distribution of the exogenous variable $R_0$ and the structural equation

$$(Z_1, ..., Z_n, R_0) \mapsto f_\gamma(Z_1, ..., Z_n, R_0) =: Y.$$

## 3.2 CAUSAL CONSISTENCY LOSS

It is not always possible to achieve an exact transformation that guarantees consistency of low- and high-level models for almost all interventions. As a consequence, we allow for the consistency between models to be approximate. To ensure that this approximation is as accurate as possible, we

minimize the expected KL divergence between the pushforward by the transformation $\tau$ of the low-level interventional distributions that we denote $\widehat{P}_\tau^{(i)}(Y, Z) = \tau_\#[P_\mathcal{L}^{(i)}(X)]$, and the corresponding interventional distribution of the high-level model $P^{(\omega(i))}$, leading to the consistency loss

$$\mathcal{L}_{\text{cons}} = \mathbb{E}_{i \sim P(i)}\left[\text{KL}\left(\widehat{P}_\tau^{(i)}(Y, Z) \| P^{(\omega(i))}(Y, Z)\right)\right]. \quad (4)$$

Other losses have been previously suggested to enforce consistency. Beckers et al. (2020) propose to take a maximum over interventions, whereas we take the expectation in our loss, thus focusing the CMR on the average performance rather than the worst case. Rischel and Weichwald (2021) and Zennaro et al. (2023) use the Jensen-Shannon (JS) divergence in the context of finite models. Instead, we choose the KL divergence because, contrary to JS, it leads to a tractable expression under Gaussian assumptions. Moreover, the proposed consistency loss (4) has the following properties.

**Proposition 3.1** (Consistency loss). *The consistency loss is positive, invariant to invertible reparametrizations (see Def. D.1), and vanishes if and only if the transformation is exact for almost all interventions. It decomposes as*

$$\mathcal{L}_{\text{cons}} = \mathbb{E}_{i \sim P(i)}\Bigg[\text{KL}\left(\widehat{P}_\tau^{(i)}(Z) \| P^{(\omega(i))}(Z)\right)$$
$$+ \mathbb{E}_{z \sim \widehat{P}_\tau^{(i)}(Z)}\left[\text{KL}\left(\widehat{P}_\tau^{(i)}(Y|z) \| P^{(0)}(Y|z)\right)\right]\Bigg], \quad (5)$$

*and is an upper bound of the* causal relevance loss

$$\mathcal{L}_{\text{rel}} = \mathbb{E}_{i \sim P(i)}\left[\text{KL}\left(\widehat{P}^{(i)}(Y) \| P^{(\omega(i))}(Y)\right)\right] \leq \mathcal{L}_{\text{cons}}. \quad (6)$$

Reparametrization invariance (see Def. D.1) refers to transformations of the pairs $(\tau, f_\gamma)$ that leave the composition $f_\gamma \circ \tau$ invariant. In the $n = 1$ linear setting (see Sec. 3.3), this corresponds to invariance by multiplicative rescaling. This guarantees that equivalent high-level causal descriptions are treated equally by the loss.

We call Eq. (5) a *Cause-Mechanism Decomposition* because the first term quantifies the *cause consistency* and the second term can be thought of as the *mechanism consistency*. This latter term assesses the similarity between the outputs of the learned high-level mechanism $P^{(0)}(Y|z)$ and the corresponding conditional distribution computed by push-forward of the low-level variables $\widehat{P}_\tau^{(i)}(Y|z)$. Since we prevent the high-level mechanism from being intervened on, only its unintervened conditional appears in the expression.

Lastly, the causal relevance loss $\mathcal{L}_{rel}$ assesses whether the variations of the target $Y$ due to low-level interventions are well-captured by high-level interventions, on average over the prior $P(i)$. Its upper bounding by $\mathcal{L}_{\text{cons}}$ ensures that by optimizing for consistency, we also indirectly promote effective "explanations" of the variations in the target density

resulting from low-level interventions. We can thus choose $P(i)$ to make the most relevant interventions more likely according to domain knowledge, such that optimizing the loss will steer towards a solution capturing the most domain-relevant variations of the target.

## 3.3 LINEAR REDUCTION WITH SHIFT INTERVENTIONS

We further constrain the setting to be able to study the solution minimizing $\mathcal{L}_{\text{cons}}$ analytically and get insights into the properties of TCR.

**Notation.** When a vector, say $\tau_k$, is associated to a high-level SCM component $k$ of a constructive transformation with alignment $\pi$, $\bar{\tau}_k$ indicates the restriction of $\tau_k$ to components in $\pi(k)$. The number of elements in a set $S$ is $\#S$.

**Tau map.** To maximize interpretability, we assume a linear $\tau$-map, represented as a vector $\tau$ such that:

$$X \mapsto \begin{bmatrix} Y \\ Z \end{bmatrix} = \begin{bmatrix} \tau_0^\top \\ \vdots, \\ \tau_n^\top \end{bmatrix} X = \begin{bmatrix} \bar{\tau}_0^\top X_{\pi(0)}, & \dots \bar{\tau}_n^\top X_{\pi(n)} \end{bmatrix}^\top.$$

**Omega map.** We focus on *shift interventions* and map the vector $i$ of low-level interventions on the nodes in $\pi(k)$ to a scalar shift intervention on the mechanism of each $Z_k$. We assume each map $\omega_k$ to be linear with vector $\omega_k$ such that

$$\omega_k(i) = \omega_k^\top i = \bar{\omega}_k^\top i_{\pi(k)}.$$

Because high-level causes are root nodes, intervening amounts to shifting the marginal distribution from $P^{(0)}(Z_k)$ to $P^{(\omega_k(i))}(Z_k) = P^{(0)}(Z_k - \omega_k(i))$.

**Choice of alignment $\pi$.** There are potential degrees of freedom for $\pi$, and users may want to incorporate domain knowledge as well as interpretability constraints to reduce the variables included in $\cup_{k \neq 0}\pi(k)$. In practice, we learn the distribution of the low-level variables among the $\pi(k)$ using regularization (see Sec. 4).

**High-level mechanism.** We use an interpretable affine high-level causal mechanism $f_\gamma$, such that

$$Y := \sum_k \alpha_k Z_k + R_0 + \beta, \quad \alpha_1, \dots, \alpha_n, \beta \in \mathbb{R}. \quad (7)$$

**Choice of prior $P(i)$.** The solutions minimizing the loss of Eq. (4), may depend on the choice of the prior $P(i)$, and in particular on which variables are actually intervened on. Let $\Omega$ denote the subset of indices of low-level variables that are intervened on with non-zero probability. The components of $i$ whose index does not belong to $\Omega$ thus take value $i = 0$ with probability one. We provide identifiability guaranties under two kinds of assumptions.

**Assumption 3.2.** $P(i_\Omega)$ has a density with respect to the Lebesgue measure, with support covering a neighborhood of zero (*i.e.* the unintervened case).

**Assumption 3.3.** The unintervened setting $i_\Omega = 0$ occurs with non-zero probability. Additionally there are at least $\#\Omega$ distinct interventions happening with non-zero probability, corresponding to a family of values of the vector of $i_\Omega$ with full rank $\#\Omega$.

While Assum. 3.2 depicts a practical setting where interventions are drawn from prior densities that reflect the prior knowledge on how likely those are, Assum. 3.3 allows addressing a classical question in causal representation learning: *How many distinct interventions are needed to learn the representation?*

## 3.4 IDENTIFIABILITY RESULTS

If we assume the low-level model is linear Gaussian of the form $X_\Omega \rightarrow X_{\pi(0)}$, we can show the existence and uniqueness of the solution.

**Proposition 3.4.** *Assume the low-level SCM follows*

$$X := AX + U + i, \quad i \sim P(i), \quad U_k \sim \mathcal{N}(\mu_k, \sigma_k^2), \sigma_k > 0,$$

*such that $X$ and $A$ take the block forms*

$$X = \begin{bmatrix} X_{\pi(0)} \\ X_\Omega \end{bmatrix}, \quad A = \begin{bmatrix} A_{00} & A_{0\Omega} \\ 0 & A_{\Omega\Omega} \end{bmatrix}.$$

*Given an arbitrary choice of linear scalar target of the form $Y = \tau_0^\top X = \bar{\tau}_0^\top X_{\pi(0)}$ and under Assum. 3.2 or Assum. 3.3, there is a unique linear 1-cause TCR (up to a multiplicative constant) satisfying $\mathcal{L}_{\text{cons}} = 0$. It is given by*

$$\pi(1) = \Omega, \quad (8)$$
$$\bar{\tau}_1 = A_{0\Omega}^\top (I_{\#\pi(0)} - A_{00})^{-\top} \bar{\tau}_0, \quad (9)$$
$$and \quad \bar{\omega}_1 = (I_{\#\Omega} - A_{\Omega\Omega})^{-\top} \bar{\tau}_1. \quad (10)$$

*Moreover, let $n_{max}$ be the maximum number $n$ such that a linear $n$-cause TCR can achieve $\mathcal{L}_{\text{cons}} = 0$. If there are no cancellations[2] among causal pathways from each node in $supp(\bar{\omega}_1)$ of Eq. (10) towards $Y$, then the $n_{max}$-cause TCR is unique up to rescaling and permutation of the causes.*

This result provides guaranties for having a unique ground-truth solution in case exact transformations can be achieved. The main assumption is the absence of feedback influences from the target set $\pi(0)$ to candidate causes. However, cycles and confounding are allowed in the low-level

---

[2]Causal pathways cancel if the linear coefficients quantifying the influence of a node on $Y$ along different directed paths of the low-level SCM sum to zero. Assuming no cancellations is akin to assuming no faithfulness violations and generically satisfied (Sprites et al., 2001, Theorem 3.2).

model, contrary to the learned high-level model. The 1-cause solution is easiest to obtain. The study of simple SCMs (App. D.2 and App. D.3) provides some insights on the form of the analytical solution. Additional results show that we lose identifiability of the TCR if we drop the assumption that not all variables in $\pi(1)$ are intervened on (see App. D.4). The $n$-cause solution is essentially a partition of the 1-cause solution that enforces independence between them.

Under the same model assumptions, the resulting constructive transformation can be associated with a constructive causal abstraction, as shown in Proposition D.3. This corresponds to a particular case of low soft abstraction introduced by Massidda et al. (2023, Def. 9).

**Example 3.5** (Linear chain). *To illustrate the solutions in Prop. 3.4, we consider a linear chain*

$$\underbrace{X_1 \to X_2 \to X_3}_{X_\Omega} \to \underbrace{X_4}_{X_{\pi(0)}}$$

*with adjacency $A_{ij} = \{1$ for $j=i+1$; $0$ else$\}$ and target $Y = X_4$, such that $\bar{\tau}_0 = I_1$. The 1-cause solution (up to a multiplicative constant) achieving $\mathcal{L}_{\text{cons}} = 0$ is*

$$\bar{\tau}_1 = \begin{pmatrix} 0 & 0 & 1 \end{pmatrix}^\top \quad and \quad \bar{\omega}_1 = \begin{pmatrix} 1 & 1 & 1 \end{pmatrix}^\top. \quad (11)$$

$\bar{\tau}_0$ *puts all its weight on the direct parent of target $X_4$ because it mediates all causal influences. In contrast, $\bar{\omega}_1$ puts weight on all variables in $\Omega$ because interventions on any of them influence $X_4$*

## 4 LINEAR TCR ALGORITHM

In this section, we introduce an algorithm to learn a linear targeted causal reduction with shift interventions.

---
**Algorithm 1** Linear TCR (LTCR)
---
**Input** $\lambda$: learning rate, $P(i)$: intervention prior, *Simulate*$(\theta, i, n_{\text{sim}})$: function returning $n_{\text{sim}}$ paths, $N_{\text{ite}}$: No. epochs, $B, B_i$: simulation/intervention batch size.
**Initialize** $\tau_1, \omega, \gamma$

  **for** $m = 1..N_{\text{ite}}$ **do**
    $X, Y \leftarrow []$
    **for** $l = 1..B_i$ **do**
      $i_l \leftarrow Sample(P(i))$
      $X_l = (x^1, .., x^B); Y_l \leftarrow Simulate(\theta, i_l, B)$
      $X \leftarrow [X[:], X_l]; Y \leftarrow [Y[:], Y_l]; I \leftarrow [I[:], i_l]$
    $L_{\text{tot}} \leftarrow ComputeLoss(X, Y, I, \tau_1, \omega_1, \gamma)$
    $\nabla_\gamma, \nabla_\tau \leftarrow ComputeLossGradient(L_{\text{tot}})$
    $(\gamma, \tau_1, \omega_1) \leftarrow (\gamma - \lambda\nabla_\gamma, \tau_1 - \lambda\nabla_{\tau_1}, \omega_1 - \lambda\nabla_{\omega_1})$
---
**Output** Estimated parameters $(\tau_1, \omega_1, \gamma)$.
---

**Gaussian approximation of consistency loss.** Since the KL divergence is challenging to compute in non-parametric settings, we make a Gaussian assumption on the densities. This allows us to obtain an analytic expression for the loss based on second order statistics (see expression in App. E.1).

**Overlap loss.** To ensure differentiability of the reduction maps we do not implement the alignment $\pi$ explicitly, but encourage non-overlapping reduction maps via the regularizer

$$\mathcal{L}_{\text{ov}} = \sum_{k<l} \left( \left\langle \frac{|\tau_k|}{\|\tau_k\|}, \frac{|\tau_l|}{\|\tau_l\|} \right\rangle + \left\langle \frac{|\omega_k|}{\|\omega_k\|}, \frac{|\omega_l|}{\|\omega_l\|} \right\rangle \right), \quad (12)$$

where $|\cdot|$ is the element-wise absolute value.

**Balancing loss.** Minimizing the Gaussian approximation of the consistency loss together with overlap regularization (12) there is nothing preventing the solution from attributing all non-zero weights in the $\tau$- and $\omega$ maps to one

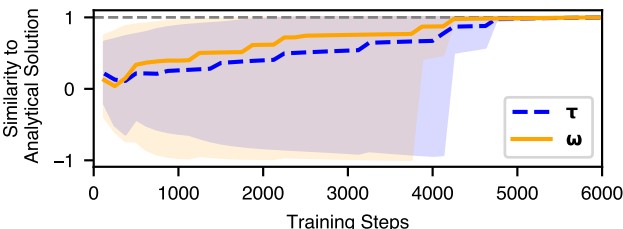

(a) **Comparison between learned and analytical solutions 1-cause TCR** Average cosine similarity to the analytical solutions over 20 runs. Each run corresponds to one draw of adjacency matrix parameters. The shaded areas show the range between the minimum and maximum values. The dashed gray line corresponds to perfect similarity.

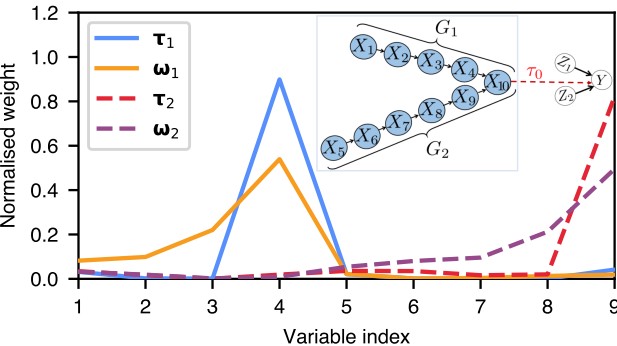

(b) **Two-branch linear model.** Learned $\tau$- and $\omega$ parameters for a TCR with two high-level variables for a linear Gaussian low-level model with $N=10$. The solid lines show the parameters for $Z_1$ and the dashed lines those for $Z_2$. The parameters are averaged over 20 runs where each run corresponds to one draw of adjacency matrix parameters. The inlay shows the causal structure of the low-level model, where two groups of variables $G_1$ and $G_2$ form two independent chains causing the target $X_{10}=Y$.

Figure 2: **Toy example experiments.**

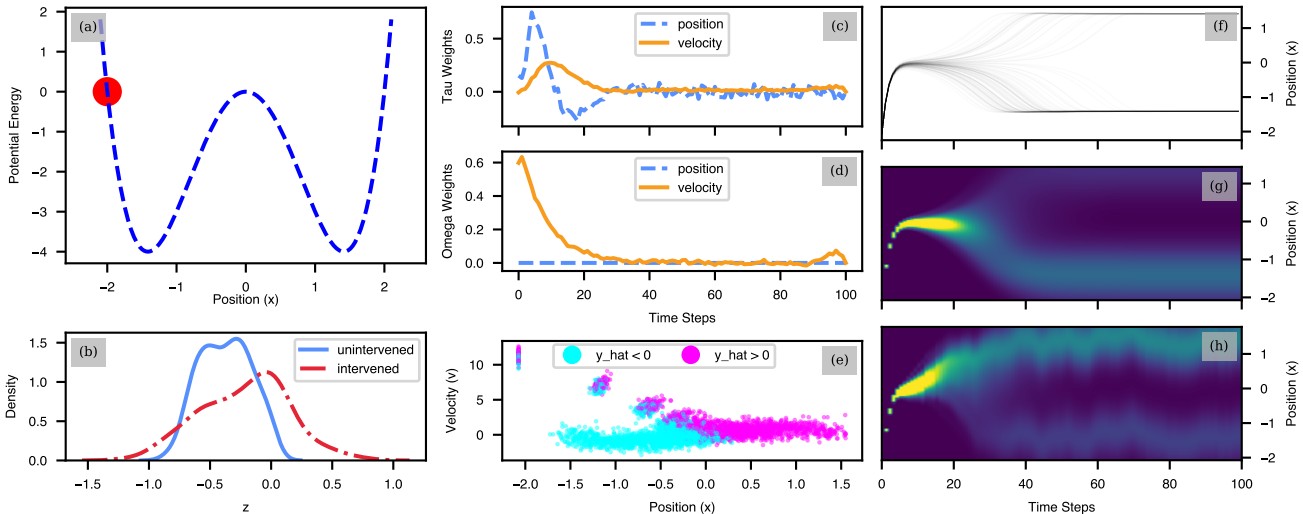

Figure 3: **Double well experiment.** (a) Experimental setup with a ball moving in a double well potential subject to linear friction. (b) Pushforward density of the high-level cause for the two settings: one where no intervention is applied (unintervened), and the other with an applied shift intervention. (c, d) Learned parameters, $\tau$ and $\omega$, respectively. The learned high-level mechanism is $f(Z_1) \approx 1.37Z_1 + 0.45$ (e) Samples in phase space (position vs. velocity) for the first 20 time points. The color indicates whether the high-level model predicts the ball to end up in the right (pink) or right well (turquoise). (f, g) Samples from the unintervened setting and the corresponding estimated density. (h) Estimated density for one intervened setting.

high-level variable while ignoring all others. In order to prevent such a collapse, we minimize stark differences between the high-level variables through the balancing term

$$\mathcal{L}_{\text{bal}} = \left( \frac{\sqrt{\sum_k \|\alpha_k \boldsymbol{\tau}_k\|_2^2}}{\sum_k \|\alpha_k \boldsymbol{\tau}_k\|_2} + \frac{\sqrt{\sum_k \|\alpha_k \boldsymbol{\omega}_k\|_2^2}}{\sum_k \|\alpha_k \boldsymbol{\omega}_k\|_2} \right), \quad (13)$$

where $\alpha_k$ is the coefficient in the linear high-level mechanism corresponding to variable $Z_k$.

Gathering the losses, we get the total objective

$$\underset{\gamma, \tau, \omega}{\text{minimize}} \ \mathcal{L}_{\text{tot}} = \mathcal{L}_{\text{cons}} + \eta_{\text{ov}}\mathcal{L}_{\text{ov}} + \eta_{\text{bal}}\mathcal{L}_{\text{bal}} . \quad (14)$$

The learning procedure is described in Algorithm 1.

# 5 EXPERIMENTS

## 5.1 TOY EXAMPLES: LINEAR GAUSSIAN LOW-LEVEL CAUSAL MODELS

**Linear low-level causal models.** We first test TCR by sampling from a linear Gaussian low-level model, rather than a simulation. We construct linear models of the form shown in Prop. 3.4 by drawing the non-zero entries in the adjacency matrix uniformly from the interval $[-1, 1]$. We learn a targeted causal reduction with two high-level variables: the target $Y$ and its single cause $Z$. Fig. 2a compares the learned $\tau_1$ and $\omega_1$ to the analytical solutions (9) and (10). We observe that, for these low-level models meeting the linear Gaussian assumption in Section 3, the learning algorithm converges to the global optimum.

**Two-branch model.** To investigate the behavior of TCR with multiple high-level variables, we consider a low-level model with two branch causal structure (Fig. 2b). With regularization for overlap (12) and balancing (13), the learned high-level variables correspond to the two branches. Within each branch, the reduction behaves as described for the linear chain in Ex. 3.5, where $\tau$ focusses on the direct parent of the target and $\omega$ is spread across all variables in the chain. Comprehensive experimental details are given in App. F.

## 5.2 DOUBLE WELL

For a simulation based on an ODE system, we learn a targeted reduction of a ball moving in a double well potential under linear friction, as shown in Fig. 3. The state vector $X$ encodes the $x$-position and velocity in $x$-direction of the ball at each time steps of the simulation. As shift-interventions, we apply small random shifts of the ball's velocity at each simulation time step, mimicking an applied external force. Initially, the ball starts on the left-hand side of the potential and starts oscillating. Since the ball experiences friction, it ends up in either the left or right minimum of the potential. The friction is relatively strong, such that, depending on the initial conditions and applied shift interventions, the ball either stays in the left well or crosses the middle hump once and stays in the right well (see Fig. 3(f)). We learn a simple TCR with a single cause $Z$ that explains the target $Y$. Further details about the nonlinear ODE system and training are given in App. F.2.

The learned TCR parameters are shown in Fig. 3(c, d). The

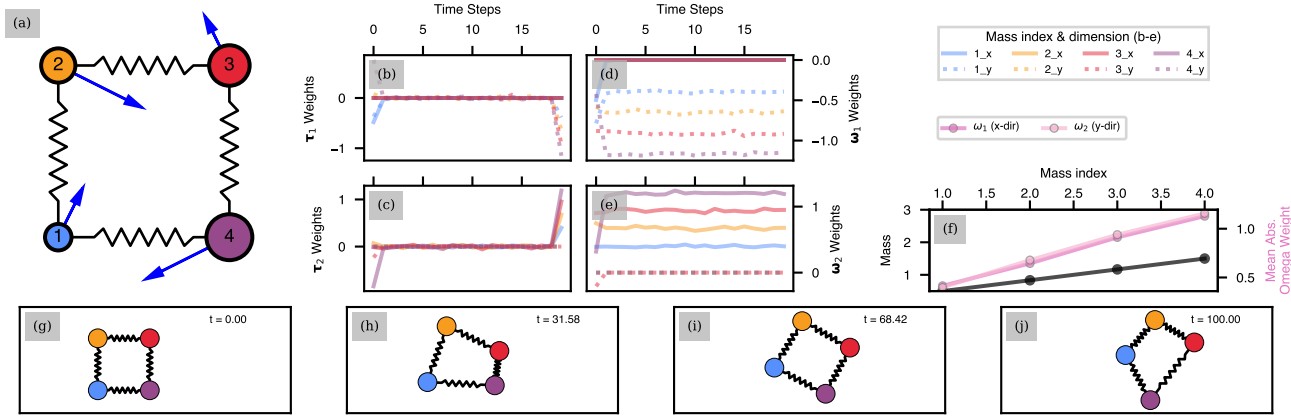

Figure 4: **Spring-mass system experiment.** (a) Simulated system of four point masses with different weights connected by springs and with random initial velocity (blue arrows). The target of the simulation is the center of mass speed in $(1, 1)$-direction. (b-e) Learned $\tau$- and $\omega$-weights corresponding to velocity components in $x$- and $y$-direction for a TCR with two high-level variables. The learned high-level mechanism is $f(Z) \approx -0.226Z_1 + 0.220Z_2$. (f) Comparison between masses and learned omega weights. For the first high-level variable the mean omega weights corresponding to the $x$-direction are shown and for the second variable those for the $y$-direction. (g-j) Example trajectory for an unintervened system.

$\tau_1$ and $\omega_1$ parameters for velocity are such that the larger the velocity is to the right, the higher $Z$ and therefore the higher the predicted target $Y$, where positive $Y$ correspond to the right well and negative to the left. Similarly, for the position parameter: the more negative the position just before the critical point of the ball crossing the hump, the higher the probability of predicting to stay in the left well. This corresponds to the correct dynamics of the system and also identifies the main drivers that influence the outcome $Y$. Fig. 3(e) shows how the learned TCR separates the phase space into simulations with enough momentum to the right to make it over the hump (pink) and those without (turquoise).

Note that TCR does not focus on the part of $X$ which best predicts the final state of the system—like the position just before the end of the simulation. It rather highlights the variables which have the most impact on the target when they are intervened on, emphasizing the decisive time span when the ball either crosses the middle hump or stays in the left well.

### 5.3 SPRING-MASS SYSTEM

We simulate a two-dimensional system of four point masses with different weights connected by springs to their respective nearest neighbors, similar to the motivating example introduced in Sec. 1. Initially, the masses are arranged in a rectangle in space such that the springs are at rest length. The masses have a random initial velocity, as shown in Fig. 4(a). As interventions, we apply random shifts to the velocities in $x$- and $y$-direction of each mass. The target of the simulation is the center of mass speed in $(1, 1)$-direction. We learn a TCR with two high-level causes. The full experimental details are given in App. F.3.

While the velocities of the individual masses are coupled,

the center of mass velocities in $x$- and $y$-direction of the system as a whole are independent, since the system is freely moving in space. The learned TCR correctly identifies these as the two independent causes of the target, with variable $Z_1$ corresponding to the $y$-direction and $Z_2$ to the $x$-direction. On average, each mass receives a similar shift in velocity through the applied interventions. However, since the masses are different, the shifts correspond to different contributions to the momentum of the system as a whole impacting the target. This is reflected in the relative weights of the learned maps being proportional to the weight of each point mass, as shown in Fig 4(f).

A second experimental setting with two groups of interconnected masses is shown in App. F.4.2, demonstrating a TCR learning independent causes along the mass index.

## 6 DISCUSSION

We introduce a novel approach for understanding complex simulations by learning high-level causal explanations from low-level models. Our Targeted Causal Reduction (TCR) framework leverages interventions to obtain simplified, high-level representations of the causes of a target phenomenon. We formulate the intervention-based consistency constraint as an information theoretic learning objective, which favors the most causally relevant explanations of the target. Under linearity and Gaussianity assumptions, we provide analytical solutions and study their uniqueness, which provides insights into TCR's governing principles. One key assumption to obtain identifiability is that the leaf node, the target, is observed. However, this is to the best of our knowledge the first identifiability proof for a general class of CMR for which the high-level variables are continuous

and partially unknown. Notably, the $n$-cause TCR provides a form of causal *independent component analysis* akin to the work of Wendong et al. (2023) but in a non-invertible setting and with a one dimensional target. We provide an algorithm for linear TCR and show it can effectively uncover the key causal factors influencing a phenomenon of interest. We demonstrate TCR on both synthetic models and scientific simulations, highlighting its potential for addressing the challenges posed by increasingly complex systems in scientific research.

While we develop a CMR framework to learn high-level explanations for simulations, the simulation itself does not have to be explicitly formulated as a causal model and the causal relationships between variables in $X$ do not have to be known a priori. The only additional element needed to learn TCR is a notion of shift-interventions. We think that most scientific simulations based on differential equations naturally allow for a reasonable notion of shift interventions.

**Limitations and future work.** To foster interpretability and tractability, we made Gaussian approximations and used linear $\tau$ and $\omega$ maps. While this has clear benefits, this may be too limiting for some complex simulations, and future work should explore more flexible approaches. Additionally, our method relies on performing a large number of interventions in simulation runs, which represents an additional cost in the context of large-scale simulation. How to make the algorithm scale to this setting is left to future work.

## Acknowledgements

We thank Sergio Hernan Garrido Mejia and Yuchen Zhu for insightful discussions.

This publication was supported by the German Federal Ministry of Education and Research (BMBF) through the Tübingen AI Center (FKZ 01IS18039A) and by the German Research Foundation (Deutsche Forschungsgemeinschaft, DFG) through the Machine Learning Cluster of Excellence (EXC 2064/1, project 390727645).

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

# Targeted Reduction of Causal Models
## (Supplementary Material)

**Armin Kekić**[1]  **Bernhard Schölkopf**[1]  **Michel Besserve**[1]

[1]Max Planck Institute for Intelligent Systems, Tübingen, Germany

## A SUPPLEMENTAL RELATED WORK

Desiderata for CMR have been addressed theoretically by several works, in particular in the context of CFL (Chalupka et al., 2015, 2016), and subsequently with the notion of *exact transformations* (Rubenstein et al., 2017) and a more strongly constrained subclass: *causal abstractions* (Beckers and Halpern, 2019). An alternative framework for composing abstractions of finite models has been proposed by Rischel and Weichwald (2021). However, only few works have addressed how to build high-level representations from the low-level system data only. A line of works focuses on language models (Geiger et al., 2021, 2023), where high-level variables and interpretations are readily available.

We start from the opposite direction and develop a general approach to build the high-level abstraction from the ground-up. Such a construction is done in CFL (Chalupka et al., 2015, 2016), where high-dimensional microscopic variables are turned into discrete high-level variables. Zennaro et al. (2023) addressed this question in the context of finite and discrete domains, by minimizing the maximum Jensen-Shannon divergence over a finite set of perfect intervention distributions. In contrast with most works on CMR, our framework is fully compatible with imperfect (soft intervention) at the low level, which are more realistic and interpretable perturbations of many real-world systems than hard interventions. Soft interventions have been used for language model alignment (Geiger et al., 2024), and their theoretical compatibility with the abstraction framework has been investigated by Massidda et al. (2023). Our approach aims at approximating an exact transformation, and is thus a relaxation of this setting.

Other theoretical frameworks for approximate abstractions have been proposed (Beckers et al., 2020; Rischel and Weichwald, 2021). Our work differs by providing an explicit loss well-suited to continuous causal models, that can be optimized efficiently and provide interpretable outcomes thanks to a cause-mechanism decomposition, a lower bound, and analytic solutions.

The way we relax constraints on low-level interventions shares also similarities with the views of Zhu et al. (2024) who consider stochastic low-level do-interventions sampled according to the observational distribution, our work is instead focused on soft-interventions, for which we impose a prior distribution, reflecting the relative importance that we put on them. Our optimization objective is averaged over this prior, such that it plays a role in the final solution.

Our approach also relates to the search for optimal interventional or counterfactual manipulations to steer the output of a system to a particular value or distribution (Amos et al., 2018; Besserve et al., 2020) or to best explain an observation (Budhathoki et al., 2022; von Kügelgen et al., 2023b). We are in a way also selecting particular manipulations, but through the choice of dimensionality reduction $\omega$, such that they are interpretable at a high level.

Finally, our approach relates to several works in causal representation learning, which have addressed identifiability of latent causal models from observational data, with (Wendong et al., 2023) or without (Squires et al., 2023; von Kügelgen et al., 2023a) assumptions on the latent causal graph. In contrast to those works, TCR does not assume an injective mapping of the mapping of the observations to the latent variables, such that the high-level model typically losses information relative to the low-level model.

# B  SUPPLEMENTAL BACKGROUND

## B.1  NUMERICAL SCHEMES FOR SIMULATIONS

Methods for the numerical approximations of scientific models is a broad area spanning multiple fields. We provide here a few elements based on a 1D example to justify how these models relate to SCMs. The Euler method (Euler, 1794), can be used to approximate a 1D ODE of the form

$$\begin{cases} x(t_0) = x_0 \,, \\ \frac{dx}{dt} = F(x(t)) \,, \end{cases}$$

with $F$ smooth real-valued function, using a discretized time grid with time step $\Delta t$. The finite difference approximation of the derivative

$$x(t + \Delta t) - x(t) \approx \Delta t \cdot \frac{dx}{dt} \,,$$

leads to the iterative numerical scheme for the approximation $\hat{x}(k\Delta t)$:

$$\begin{cases} \hat{x}(t_0) = x_0 \,, \\ \hat{x}(t_0 + (k+1)\Delta t) = \hat{x}(t_0 + k\Delta t) + \Delta t \cdot F(\hat{x}(t_0 + k\Delta t)) \,, \quad k > 0 \,. \end{cases}$$

This scheme is called *explicit* because future values depend explicitly on past ones. In that case, one may define the $N$ low-level endogenous variables as $X = [\hat{x}(t_0 + \Delta t), \ldots, \hat{x}(t_0 + N\Delta t)]^\top$. They can be seen as pertaining to a chain SCM with structural equations

$$\begin{cases} \widehat{X}_1 := x_0 + \Delta t \cdot F(x_0) \\ \widehat{X}_{k+1} := \widehat{X}_k + \Delta t \cdot F(\widehat{X}_k) \,, \quad k > 1 \,. \end{cases}$$

Because the ODE describes deterministic dynamics, the corresponding SCM is deterministic as well, *i.e.* exogenous variables can be taken as trivial zero constants. However, if we turn this ODE into the following 1D SDE

$$\begin{cases} X(t_0) = x_0 \,, \\ dX = F(X(t))dt + \sigma_W \cdot dW \,, \end{cases}$$

where $W$ is a standard Brownian motion, then the Euler-Murayama method generalizes the previous approximation (Sauer, 2013), and leads to an updated SCM with structural equation

$$\begin{cases} \widehat{X}_1 := x_0 + \Delta t \cdot F(x_0) + U_1 \\ \widehat{X}_{k+1} := \widehat{X}_k + \Delta t \cdot F(\widehat{X}_k) + U_{k+1} \,, \quad k > 1 \,, \end{cases}$$

where the exogenous variables $U_k$ represent the increments of the scaled Brownian motion $\sigma_W \cdot W$ between successive time steps, and are thus jointly independent Gaussian due to fundamental properties of Brownian motion.

This approach generalizes to explicit numerical schemes for multivariate ODE and SDEs where the state variable $X$ is an element of $\mathbb{R}^n$. As an illustration, we can take the following class of SDE models

$$\begin{cases} X(t_0) = x_0 \,, \\ dX = F(X(t))dt + \sigma_W \cdot dW \,, \end{cases}$$

with $F : \mathbb{R}^n \to \mathbb{R}^n$, $\sigma_W = \mathbb{R}^{(n \times n)}$, and $W$ a $n$-dimensional standard Brownian motion. This leads to the scheme

$$\begin{cases} \widehat{X}_1 := x_0 + \Delta t \cdot F(x_0) + U_1 \\ \widehat{X}_{k+1} := \widehat{X}_k + \Delta t \cdot F(\widehat{X}_k) + U_{k+1} \,, \quad k > 1 \,, \end{cases}$$

where the $U_k$ are now multivariate Gaussian variables, whose components may or may not be independent depending on the choice of the matrix $\sigma_W$. If the exogenous components are independent, the variables can be described by a standard SCM as introduced in the main text. If the exogenous components are dependent, the variables can be described by a more general notion of SCM, allowing hidden confounding (Bongers et al., 2021).

Further generalization to numerical schemes for Stochastic Partial Differential Equations (SPDEs) using finite difference approximations for partial derivatives with respect to other variables than time are also possible (Millet and Morien, 2005).

## B.2 REDUCTION OF THE EULER SCHEME FOR A SYSTEM OF POINT MASSES

In the context of the main text example, we assume each point mass is submitted to a fluid friction force opposing its movement with fixed coefficient $\lambda$. Masses are moreover intervened on via additional external forces $\{f_k\}$. Finally, internal forces are exerted on mass $k$ by other point masses of the system, summing up to $g_k$. Newton's second law applied to individual masses results in the following system of 2D vector equations

$$m_k \frac{d\boldsymbol{v}_k}{dt} = -\lambda \boldsymbol{v}_k(t) + \boldsymbol{f}_k(t) + \boldsymbol{g}_k(t) \,.$$

We can approximate each equation to iteratively estimate the $x$ and $y$ components of the speed of individual point masses in the system, using a small time-step $\Delta t$, such that we get the discrete time estimates $\hat{v}_{x,k}[n] \approx v_{x,k}(n\Delta t)$ and $\hat{v}_{y,k}[n] \approx v_{y,k}(n\Delta t)$ satisfying

$$m_k \cdot \hat{v}_{x,k}[n+1] := (1 - \Delta t\lambda) \cdot m_k \cdot \hat{v}_{x,k}[n] + \Delta t \cdot f_{x,k}(n\Delta t) + \Delta t \cdot g_{x,k}(n\Delta t) \,,$$
$$m_k \cdot \hat{v}_{y,k}[n+1] := (1 - \Delta t\lambda) \cdot m_k \cdot \hat{v}_{y,k}[n] + \Delta t \cdot f_{y,k}(n\Delta t) + \Delta t \cdot g_{y,k}(n\Delta t) .$$

Here, $f$ represents external forces, $\lambda$ is a viscous damping coefficient, and $g$ denotes internal forces. We consider $\boldsymbol{i}$ to be the vector of all components of external forces, and the target variable to be the final horizontal speed of the center of mass at iteration $N$. From the physics of freely moving systems of points, it is clear that the target variable can be predicted by considering only the horizontal dynamics of the center of mass. More precisely, we integrate the sum of external forces over the time span of the experiment, and use the last intervened time point $n_f$ to predict the final outcome of the simulation, leading to the reduction

$$Z_1^{(\omega(\boldsymbol{i}))} = \left(\sum_k m_k\right) v_{x,G}[n_f] = \sum_k m_k v_{x,k}[n_f] = \Delta t \sum_{n=0}^{n_f} (1 - \Delta t\lambda)^{(n_f - n)} \sum_k f_{x,k}(n\Delta t) \,,$$

$$Y = v_{x,G}[N] := (1 - \Delta t\lambda)^{(N-n_f)} Z_1 + \sum_{n=n_f+1}^{N} \sum_k f_{x,k}(n\Delta t) \,.$$

To make the notation compatible with that used in our TCR framework, we can gather all speed variables in a high-dimensional vector $\boldsymbol{X}$ and all external force variables in a vector $\boldsymbol{i}$, the high-level causal model is thus generated by a linear $\tau$-map and linear $\omega$ map for shift interventions, taking the form of the exact transformation

$$Z_1 = \boldsymbol{\tau}_1^\top \boldsymbol{X} + \boldsymbol{\omega}_1^\top \boldsymbol{i} \,, \quad Y = \boldsymbol{\tau}_0^\top \boldsymbol{X} + \boldsymbol{\omega}_0^\top \boldsymbol{i} := f(Z_1) \,.$$

where the term $\boldsymbol{\omega}_0^\top \boldsymbol{i}$ accounts for interventions happening between discrete times $n_f + 1$ and $N$ and thus affect $Y$ without being mediated by $Z_1$. In our framework, only interventions mediated by the cause, reflected in the term $\boldsymbol{\omega}_1^\top \boldsymbol{i}$, are accounted for in the high-level model.

## B.3 CONSTRUCTIVE TRANSFORMATIONS

We complete the main text definition to include the constraint on the intervention map $\omega$

**Definition B.1.** $(\tau, \omega) : (\mathcal{X} \to \mathcal{Z}, \mathcal{I} \to \mathcal{J})$ *is a constructive $(\tau - \omega)$-transformation between model $\mathcal{L}$ and $\mathcal{H}$ if there exists an alignment map $\pi$ mapping each high-level endogenous variable to a subset of low-level endogenous variables such that for all $k \neq l$, $\pi(k) \cap \pi(n) = \emptyset$ and we have both*

- *for each component $\tau_k$ of $\tau$ there exists a function $\bar{\tau}_k$ such that for all $\boldsymbol{x}$ in $\mathcal{X}$,*

$$\tau_k(\boldsymbol{x}) = \bar{\tau}_k(\boldsymbol{x}_{\pi(k)}) \,;$$

- *for each component $\omega_k$ of $\omega$ there exists a function $\bar{\omega}_k$ such that for all $\boldsymbol{i}$ in $\mathcal{I}$,*

$$\omega_k(\boldsymbol{i}) = \bar{\omega}_k(\boldsymbol{i}_{\pi(k)}) \,.$$

# C PROOF OF MAIN TEXT RESULTS

## C.1 PROOF OF PROPOSITION 3.1

We first reformulate Proposition 3.1 more formally as follows.

**Proposition C.1.** *The consistency loss is positive, invariant to invertible reparametrizations as defined in Definition D.1, and vanishes if and only if the transformation is exact for almost all interventions. It admits the following decomposition:*

$$\mathcal{L}_{\text{cons}} = \mathbb{E}_{i \sim P(i)} \left[ \text{KL}\left( \widehat{P}_\tau^{(i)}(\mathbf{Z}) \, || P^{(\omega(i))}(\mathbf{Z}) \right) + \mathbb{E}_{z \sim \widehat{P}_\tau^{(i)}(\mathbf{Z})} \left[ \text{KL}\left( \widehat{P}_\tau^{(i)}(Y|z) \right) || P^{(0)}(Y|z) \right) \right] \right], \tag{15}$$

*and is an upper bound of the* causal relevance loss

$$\mathcal{L}_{\text{rel}} = \mathbb{E}_{i \sim P(i)} \left[ \text{KL}\left( \widehat{P}^{(i)}(Y) \, || P^{(\omega(i))}(Y) \right) \right] \leq \mathcal{L}_{\text{cons}}. \tag{16}$$

*Proof.* **Positivity** of the loss comes from the positivity of the KL-divergence. Taking the expectation of this divergence with respect to $P(i)$ thus must be positive too.

**Invariance to reparameterizations.** We assume a reparametrization $\rho$ designed according to the framework introduced in Appendix D.1. By invariance of the KL divergence to invertible transformations, we have equality between the KL associated to the two different reductions $(\tau, \omega)$ and $(\rho \circ \tau, \psi \circ \omega)$:

$$\text{KL}\left( \widehat{P}_\tau^{(i)}(Y, \mathbf{Z}) || P_{\mathcal{H},\gamma}^{(\omega(i))}(Y, \mathbf{Z}) \right) = \text{KL}\left( \tilde{\rho}_\#[\widehat{P}_\tau^{(i)}(Y, \mathbf{Z})] || \tilde{\rho}_\#[P_{\mathcal{H},\gamma}^{(\omega(i))}(Y, \mathbf{Z})] \right) = \text{KL}\left( \widehat{P}_{\rho \circ \tau}^{(i)}(Y, \mathbf{Z}) || P_{\mathcal{H},\gamma'}^{(\psi \circ \omega(i))}(Y, \mathbf{Z}) \right).$$

The transformation $(\rho, \psi)$ thus leaves $\mathcal{L}_{\text{cons}}$ invariant.

**Cause-mechanism decomposition.** Under our setting (see Sec. 3.1), the interventional distribution of the high-level causal model factorizes as

$$P^{(\omega(i))}(Y, \mathbf{Z}) = P^{(0)}(Y|\mathbf{Z}) P^{(\omega(i))}(\mathbf{Z}).$$

The pushforward (by reduction) of the interventional distribution of the low-level model factorizes as

$$\widehat{P}^{(i)}(Y, \mathbf{Z}) = \widehat{P}^{(i)}(Y|\mathbf{Z}) \widehat{P}^{(i)}(\mathbf{Z}),$$

with $\widehat{P}^{(i)}(\mathbf{Z}) = \tau_{1,\#}[P^{(i)}(X_{\pi(1)})]$ and $\widehat{P}^{(i)}(Y|\mathbf{Z}) = \frac{\tau_\#\left[ P^{(i)}(X_{\pi(0)}, X_{\pi(1)}) \right]}{\tau_{1,\#}[P^{(i)}(X_{\pi(1)})]}$.

Thus, the KL divergence can be decomposed as

$$\begin{aligned}
\text{KL}&\left( \widehat{P}^{(i)}(Y, \mathbf{Z}) || P^{(\omega(i))}(Y, \mathbf{Z}) \right) \\
&= \int_{\mathcal{Y}} \int_{\mathcal{Z}} \widehat{P}^{(i)}(Y, \mathbf{Z}) \log \frac{\widehat{P}^{(i)}(Y, \mathbf{Z})}{P^{(\omega(i))}(Y, \mathbf{Z})} d\mathbf{Z} dY \\
&= \int_{\mathcal{Y}} \int_{\mathcal{Z}} \widehat{P}^{(i)}(Y|\mathbf{Z}) \widehat{P}^{(i)}(\mathbf{Z}) \log \frac{\widehat{P}^{(i)}(Y|\mathbf{Z}) \widehat{P}^{(i)}(\mathbf{Z})}{P^{(0)}(Y|\mathbf{Z}) P^{(\omega(i))}(\mathbf{Z})} d\mathbf{Z} dY \\
&= \text{KL}_{\mathbf{Z}}\left( \widehat{P}^{(i)}(\mathbf{Z}) || P^{(\omega(i))}(\mathbf{Z}) \right) + \mathbb{E}_{z \sim \widehat{P}^{(i)}(\mathbf{Z})} \left[ \text{KL}_Y\left( \widehat{P}^{(i)}(Y|\mathbf{Z} = z) || P^{(0)}(Y|\mathbf{Z} = z) \right) \right] \\
&= \text{KL}_{\mathbf{Z}}\left( \widehat{P}^{(i)}(\mathbf{Z}) || P^{(\omega(i))}(\mathbf{Z}) \right) + \text{KL}_{Y,\mathbf{Z}}\left( \widehat{P}^{(i)}(\widehat{Y}, \mathbf{Z}) || P^{(0)}(Y|\mathbf{Z}) \widehat{P}^{(i)}(\mathbf{Z}) \right).
\end{aligned}$$

We call the first term *cause consistency loss*, as it matches the definition of a consistency loss but for cause variables only. The second term can be thought of as a *mechanism consistency loss*, where we use the ground truth low-level cause distribution to probe the similarity of the outputs of the "true" (in fact, the conditional distribution) and approximate mechanism. Our interpretability choice prevents the high-level mechanism from being intervened on, so a single stochastic map (*i.e.* a Markov kernel) must fit at best all the sampled experimental conditionals.

**Lower bounding by causal relevance**   We may ask the question of causal relevance of high-level causes. One way to quantify this is to assess whether the variations of the target due to low-level interventions are well captured by high-level interventions, which can be measured by a KL divergence on the target's marginal

$$\mathcal{L}_{\text{rel}} = \mathbb{E}_{\boldsymbol{i} \sim p(\boldsymbol{i})} \left[ \text{KL}_Y \left( \widehat{P}^{(\boldsymbol{i})}(Y) \,||\, P^{(\omega(\boldsymbol{i}))}(Y) \right) \right] \,.$$

Note: In the Gaussian 1D case, the formula for the causal relevance loss is

$$\mathcal{L}_{\text{rel}} = \frac{1}{2} \mathbb{E}_{\boldsymbol{i} \sim p(\boldsymbol{i})} \left[ \frac{(\mu_Y + \alpha \boldsymbol{\omega}^\top \boldsymbol{i} - \widehat{\mu}_Y^{(\boldsymbol{i})})^2}{\sigma_Y^2} + \frac{\widehat{\sigma^2}_Y^{(\boldsymbol{i})}}{\sigma_Y^2} - \ln \left( \frac{\widehat{\sigma^2}_Y^{(\boldsymbol{i})}}{\sigma_Y^2} \right) - 1 \right] \,.$$

Interestingly, we can break down this term using

$$\text{KL}\left( \widehat{P}^{(\boldsymbol{i})}(Y, \boldsymbol{Z}) \,||\, P^{(\omega(\boldsymbol{i}))}(Y, \boldsymbol{Z}) \right)$$
$$= \text{KL}_Y \left( \widehat{P}^{(\boldsymbol{i})}(Y) \,||\, P^{(\omega(\boldsymbol{i}))}(Y) \right) + \mathbb{E}_{y \sim \widehat{P}^{(\boldsymbol{i})}(Y)} \left[ \text{KL}_Y \left( \widehat{P}^{(\boldsymbol{i})}(\boldsymbol{Z}|Y = y) \,||\, P^{(\omega(\boldsymbol{i}))}(\boldsymbol{Z}|Y = y) \right) \right]$$

where both terms are positive by positivity of the KL divergence. As a consequence,

$$\text{KL}_Y \left( \widehat{P}^{(\boldsymbol{i})}(Y) \,||\, P^{(\omega(\boldsymbol{i}))}(Y) \right) = \text{KL}\left( \widehat{P}^{(\boldsymbol{i})}(Y, \boldsymbol{Z}) \,||\, P^{(\omega(\boldsymbol{i}))}(Y, \boldsymbol{Z}) \right) - \mathbb{E}_{y \sim \widehat{P}^{(\boldsymbol{i})}(Y)} \left[ \text{KL}_Y \left( \widehat{P}^{(\boldsymbol{i})}(\boldsymbol{Z}|Y = y) \,||\, P^{(\omega(\boldsymbol{i}))}(\boldsymbol{Z}|Y = y) \right) \right]$$
$$\leq \text{KL}\left( \widehat{P}^{(\boldsymbol{i})}(\widehat{Y}, \boldsymbol{Z}) \,||\, P^{(\omega(\boldsymbol{i}))}(Y, \boldsymbol{Z}) \right) = \mathcal{L}_{\text{cons}} \,.$$

so the minimized consistency loss is an upper bound to causal relevance.

$$\square$$

## C.2   PROOF OF PROPOSITION 3.4

**Proposition 3.4.** *Assume the low-level SCM follows*

$$\boldsymbol{X} := A\boldsymbol{X} + \boldsymbol{U} + \boldsymbol{i} \,, \quad \boldsymbol{i} \sim P(\boldsymbol{i}) \,, \quad U_k \sim \mathcal{N}(\mu_k, \sigma_k^2) \,, \sigma_k > 0 \,,$$

*such that $\boldsymbol{X}$ and $A$ take the block forms*

$$\boldsymbol{X} = \begin{bmatrix} \boldsymbol{X}_{\pi(0)} \\ \boldsymbol{X}_\Omega \end{bmatrix} \,, \quad A = \begin{bmatrix} A_{00} & A_{0\Omega} \\ \boldsymbol{0} & A_{\Omega\Omega} \end{bmatrix} \,.$$

*Given an arbitrary choice of linear scalar target of the form $Y = \boldsymbol{\tau}_0^\top \boldsymbol{X} = \bar{\boldsymbol{\tau}}_0^\top \boldsymbol{X}_{\pi(0)}$ and under Assum. 3.2 or Assum. 3.3, there is a unique linear 1-cause TCR (up to a multiplicative constant) satisfying $\mathcal{L}_{\text{cons}} = 0$. It is given by*

$$\pi(1) = \Omega \,, \tag{8}$$
$$\bar{\boldsymbol{\tau}}_1 = A_{0\Omega}^\top (I_{\#\pi(0)} - A_{00})^{-\top} \bar{\boldsymbol{\tau}}_0 \,, \tag{9}$$
$$\text{and} \quad \bar{\boldsymbol{\omega}}_1 = (I_{\#\Omega} - A_{\Omega\Omega})^{-\top} \bar{\boldsymbol{\tau}}_1 \,. \tag{10}$$

*Moreover, let $n_{max}$ be the maximum number $n$ such that a linear $n$-cause TCR can achieve $\mathcal{L}_{\text{cons}} = 0$. If there are no cancellations[1] among causal pathways from each node in $\text{supp}(\bar{\boldsymbol{\omega}}_1)$ of Eq. (10) towards $Y$, then the $n_{max}$-cause TCR is unique up to rescaling and permutation of the causes.*

---

[1]Causal pathways cancel if the linear coefficients quantifying the influence of a node on $Y$ along different directed paths of the low-level SCM sum to zero. Assuming no cancellations is akin to assuming no faithfulness violations and generically satisfied (Sprites et al., 2001, Theorem 3.2).

*Proof.* **Part 1: 1-cause TCR.**

**Overview.** We exploit the positive definiteness of the KL loss and its continuity with respect to $i$. Since the variables are jointly Gaussian, continuity is obvious from the analytical expression of the KL for Gaussian variables and continuity of the shift operation applied to the parameters of the Gaussian. We exploit the cause-mechanism decomposition and the lower-bound by $\mathcal{L}_{\text{cons}}$ to progressively identify necessary conditions on parameters to have $\mathcal{L}_{\text{cons}} = 0$ and finally check those conditions are sufficient.

**Preliminaries.** Let $N_0$ denote the size of $\pi(0)$ and $N_1$ be the size of $\pi(1)$. The SCM is assumed uniquely solvable (Definition 2.1), such that $x = Ax + u$ has a unique solution for almost all values of $u$. Since $U$ has full support, this implies that $I_N - A$ is invertible. The low-level variables then satisfy

$$X^{(i)} = (I_N - A)^{-1}(U + i)$$

where, due to the block triangular form of $A$,

$$(I_N - A)^{-1} = \begin{bmatrix} (I_{N_0} - A_{00})^{-1}, & (I_{N_0} - A_{00})^{-1}A_{0\Omega}(I_{N_1} - A_{\Omega\Omega})^{-1} \\ 0 & (I_{N_1} - A_{\Omega\Omega})^{-1}. \end{bmatrix}$$

In the assumed model all low-level variables are either in $\pi(0)$ or in $\Omega$. Since the CMR is constructive we have $\pi(1) \subset \Omega$. Without loss of generality, we can impose $\pi(1) = \Omega$ by setting the unused components of $\tau_1$ and $\omega_1$ to zero. For an arbitrary interventional setting $i$, this leads to the mapping of the low-level variable to the high-level cause variable, which we denote $\widehat{Z}_1^{(i)} = \tau_1(X^{(i)})$, to satisfy

$$\widehat{Z}_1^{(i)} = \tau_1^\top (I_N - A)^{-1}(U + i) = \bar{\tau}_1^\top (I_{N_1} - A_{\Omega\Omega})^{-1}(U_\Omega + i_\Omega). \tag{17}$$

Moreover, because we assume also shift interventions in the high-level model, the cause $Z_1$ in this model has an interventional distribution satisfying

$$P^{(\omega_1(i))}(Z_1) = P^{(0)}\left(Z_1 - \omega_1^\top(i)\right).$$

**Necessary conditions.** We are looking for solutions satisfying $\mathcal{L}_{\text{cons}} = 0$. By positivity of the KL divergence, this implies that for almost all $i$, the distribution of $\widehat{Z}_1^{(i)}$ matches the learned high-level interventional distribution of high-level cause $Z_1$, which satisfies

$$P^{(\omega_1(i))}(Z_1) = P^{(0)}\left(Z_1 - \omega_1^\top(i)\right).$$

**Matching unintervened distributions of $Z_1$.** If Assum. 3.2 holds, the prior $P(i_\Omega)$ has density with respect to the Lebesgue measure with support including a neighborhood of $i_\Omega = 0$. By continuity of the KL divergence with respect to the intervention parameters, a solution making the consistency loss vanish needs to have the KL divergence term vanish for $i_\Omega = 0$ (otherwise we could find a neighborhood of $i_\Omega = 0$ such that the KL divergence does not vanish, by continuity of the KL divergence, and $\mathcal{L}_{\text{cons}}$ would be non-vanishing, contradicting our assumption).

Alternatively, Assum. 3.3 also obviously implies vanishing of the KL divergence for the unintervened setting.

This vanishing of the KL divergence entails, again by positivity, that its terms, the two unintervened densities, are equal, such that we get, using Eq. (17)

$$P^{(0)}(Z_1) = (\tau_1)_\#\left[P(X)\right] = (\bar{\tau}_1^\top(I_{N_1} - A_{\Omega\Omega})^{-1})_\#\left[P(U_\Omega)\right] \tag{18}$$

$$= \mathcal{N}(\bar{\tau}_1^\top(I_{N_1} - A_{\Omega\Omega})^{-1}\mu_\Omega, \bar{\tau}_1^\top(I_{N_1} - A_{\Omega\Omega})^{-1}\Sigma_\Omega(I_{N_1} - A_{\Omega\Omega})^{-\top}\bar{\tau}_1), \tag{19}$$

which entails the following constraints on the variance and mean of the high-level cause

$$\sigma_{Z_1}^2 = \bar{\tau}_1^\top(I_{N_1} - A_{\Omega\Omega})^{-1}\Sigma_\Omega(I_{N_1} - A_{\Omega\Omega})^{-\top}\bar{\tau}_1 \tag{20}$$

and

$$\mu_{Z_1} = \bar{\tau}_1^\top(I_{N_1} - A_{\Omega\Omega})^{-1}\mu_\Omega. \tag{21}$$

**Matching interventional distributions of $Z_1$.** For the same reasons, under Assum. 3.2, we can further match the interventional distributions in an open set included in the interior of the support of $P(i)$, such that for all $i$ in this open set the

following distributions are the same

$$P^{(\omega_1(i))}(Z_1) = \mathcal{N}(\bar{\tau}_1^\top (I_{N_1} - A_{\Omega\Omega})^{-1}(\mu_\Omega + i_\Omega), \bar{\tau}_1^\top (I_{N_1} - A_{\Omega\Omega})^{-1}\Sigma_\Omega (I_{N_1} - A_{\Omega\Omega})^{-\top}\bar{\tau}_1) \text{ and}$$

$$(\tau_1)_\# \left[ P^{(i)}(X) \right] = \mathcal{N}(\mu_{Z_1} + \omega_1^\top i, \sigma_{Z_1}^2) = \mathcal{N}(\bar{\tau}_1^\top (I_{N_1} - A_{\Omega\Omega})^{-1}\mu_\Omega + \omega_1^\top i, \sigma_{Z_1}^2).$$

Indeed, otherwise the KL would not vanish in a neighborhood of non-zero measure and would contradict the assumption that $\mathcal{L}_{\text{cons}}$ vanishes.

This implies that for all $i$ in this open neighborhood

$$\bar{\tau}_1^\top (I_{N_1} - A_{\Omega\Omega})^{-1}(\mu_\Omega + i_\Omega) = \bar{\tau}_1^\top (I_{N_1} - A_{\Omega\Omega})^{-1}\mu_\Omega + \bar{\omega}_1^\top i_\Omega,$$

which simplifies to

$$\bar{\tau}_1^\top (I_{N_1} - A_{\Omega\Omega})^{-1}i_\Omega = \bar{\omega}_1^\top i_\Omega.$$

Since this equality between two linear functions of $i_{\pi(1)}$ is valid on an open set of the vector space of $i_{\pi(1)}$, these functions must be equal (we can reparameterize $i$ to show that the linear maps must match on a basis of the space, so they are equal). This is valid if and only if, in addition to Eqs. (20-21),

$$\bar{\omega}_1 = (I_{N_1} - A_{\Omega\Omega})^{-\top}\bar{\tau}_1, \tag{22}$$

is verified.

Alternatively, we obtain the same result by replacing Assum. 3.2 by Assum. 3.3. Indeed, the finite distribution over interventions imposes that the KL term inside the expectation must vanish for each of them (including the unintervened distribution). As long as the collection of finite interventions vectors forms a rank $\#\Omega = \#\pi(1) = N_1$ family, we can choose a subset of $N_1$ such vectors $\{i_\Omega^1, \ldots, i_\Omega^{N_1}\}$ such that it forms a linearly independent family. It can be used to build the matrix equality

$$\bar{\tau}_1^\top (I_{N_1} - A_{\Omega\Omega})^{-1} \left[ i_\Omega^1, \ldots, i_\Omega^{N_1} \right] = \omega_1^\top \left[ i_\Omega^1, \ldots, i_\Omega^{N_1} \right] \tag{23}$$

where the matrix $\left[ i_\Omega^1, \ldots, i_\Omega^{N_1} \right]$ is invertible. By right-multiplying Eq. (23) by this inverse, we obtain Eq. (22) again.

**Matching distributions of $Y$.** We can move on to check the implication of consistency of the effect's conditional. It entails for almost all of $i$

$$\widehat{P}^{(i)}(Y|Z)\widehat{P}^{(i)}(Z) = P^{(\omega(i))}(Y|Z)\widehat{P}^{(i)}(Z) = P^{(0)}(Y|Z)\widehat{P}^{(i)}(Z).$$

Introducing $T = \begin{bmatrix} \tau_0^\top \\ \tau_1^\top \end{bmatrix}$ the left-hand side is obtained by using

$$(Y, Z) \sim \mathcal{N}(T\mu_X, T\Sigma_X T^\top).$$

And the right-hand side by using

$$Y = f(Z_1) + R_0.$$

Fitting first only the marginals of $Y$, we obtain necessary conditions. We have

$$\widehat{P}^{(i)}(Y) = P^{(\omega(i))}(Y)$$

where for the left-hand side

$$Y = \bar{\tau}_0^\top (I_{N_0} - A_{00})^{-1}U_{\pi(0)} + \bar{\tau}_0^\top (I_{N_0} - A_{00})^{-1}A_{0\Omega}(I_{N_1} - A_{\Omega\Omega})^{-1}(U_{\pi(1)} + i)$$

and for the right-hand side

$$Y \sim f_\# [P^{(\omega(i))}(Z_1)] * P(R_0).$$

Given the affine mechanism assumption of Eq. (7), $f(Z) = \alpha Z + \beta$ and under Assum. 3.2, equality of marginal distributions entails the following equality for all $i_\Omega$ in an open neighborhood of 0 (otherwise $\mathcal{L}_{\text{rel}} \leq \mathcal{L}_{\text{cons}}$ would not vanish)

$$\bar{\tau}_0^\top (I_{N_0} - A_{00})^{-1}\mu_{\pi(0)} + \bar{\tau}_0^\top (I_{N_0} - A_{00})^{-1}A_{0\Omega}(I_{N_1} - A_{\Omega\Omega})^{-1}(\mu_{U_1} + i_\Omega) = \alpha\bar{\tau}_1^\top (I_{N_1} - A_{\Omega\Omega})^{-1}(\mu_\Omega + i_\Omega) + \beta + \mu_{R_0}$$

which requires (setting $i = 0$)

$$\beta + \mu_{R_0} = \bar{\tau}_0^\top (I_{N_0} - A_{00})^{-1} \mu_{\pi(0)} .$$

We can fix $\mu_{R_0}$ to zero to avoid redundancy of additive constants, such that

$$\mu_{Y|Z=0} = \beta = \bar{\tau}_0^\top (I_{N_0} - A_{00})^{-1} \mu_{\pi(0)} \tag{24}$$

and consistency of non-zero shift interventions additionally entail for all $i_\Omega$ in the support of $P(i_\Omega)$

$$\bar{\tau}_0^\top (I_{N_0} - A_{00})^{-1} A_{0\Omega} (I_{N_1} - A_{\Omega\Omega})^{-1} i_\Omega = \alpha \bar{\tau}_1^\top (I_{N_1} - A_{\Omega\Omega})^{-1} i_\Omega .$$

Since one can always choose a linearly independent family of vectors $i_\Omega$ within the open neighborhood of zero for which this equality holds, this yields

$$\bar{\tau}_0^\top (I_{N_0} - A_{00})^{-1} A_{0\Omega} (I_{N_1} - A_{\Omega\Omega})^{-1} = \alpha \bar{\tau}_1^\top (I_{N_1} - A_{\Omega\Omega})^{-1} .$$

Then, right-multiplying by $(I_{N_1} - A_{\Omega\Omega})$, we get

$$A_{0\Omega}^\top (I_{N_0} - A_{00})^{-\top} \bar{\tau}_0 = \alpha \bar{\tau}_1 . \tag{25}$$

Similarly as above, the same conclusion can be drawn if we replace Assum. 3.2 by Assum. 3.3.

**Sufficiency of the constraints.** We have derived the expressions of the TCR parameters in Eqs. (22,24,25) from necessary conditions for matching the marginals of high-level variables, $P(Z_1)$ and $P(Y)$, to their corresponding pushforward distributions of the low-level variables, $(\tau_1)_\#[P(X)]$ and $(\tau_0)_\#[P(X)]$. Now what remains is to check the same for conditional distributions to show those conditions are sufficient. Indeed, this implies that the joint high-level distributions of $(Y, Z_1)$ and $(\tau_0(X), \tau_1(X))$ are matching.

Let us first note that given that the low-level model, the $\tau$ maps and the high-level mechanisms are linear or affine, and that the low-level exogenous variables are Gaussian, the exogenous high-level variable will necessarily be Gaussian as well to satisfy the consistency constraints.

Let us now compute the covariance matrix of the low-level variables.

$$cov(X) = \begin{bmatrix} (I_{N_0} - A_{00})^{-1} & (I_{N_0} - A_{00})^{-1} A_{0\Omega} (I_{N_1} - A_{\Omega\Omega})^{-1} \\ \mathbf{0} & (I_{N_1} - A_{\Omega\Omega})^{-1} \end{bmatrix} \Sigma_U \begin{bmatrix} (I_{N_0} - A_{00})^{-\top} & \mathbf{0} \\ (I_{N_1} - A_{\Omega\Omega})^{-\top} A_{0\Omega}^\top (I_{N_0} - A_{00})^{-\top} & (I_{N_1} - A_{\Omega\Omega})^{-\top} \end{bmatrix} .$$

Because the exogenous covariance is block diagonal, we get

$$= \begin{bmatrix} (I_{N_0} - A_{00})^{-1} A_{0\Omega} (I_{N_1} - A_{\Omega\Omega})^{-1} \Sigma_\Omega (I_{N_1} - A_{\Omega\Omega})^{-\top} A_{0\Omega}^\top (I_{N_0} - A_{00})^{-\top} & (I_{N_0} - A_{00})^{-1} A_{0\Omega} (I_{N_1} - A_{\Omega\Omega})^{-1} \Sigma_\Omega (I_{N_1} - A_{\Omega\Omega})^{-\top} \\ + (I_{N_0} - A_{00})^{-1} \Sigma_{\pi(0)} (I_{N_0} - A_{00})^{-\top} & \\ (I_{N_1} - A_{\Omega\Omega})^{-1} \Sigma_\Omega (I_{N_1} - A_{\Omega\Omega})^{-\top} A_{0\Omega}^\top (I_{N_0} - A_{00})^{-\top} & (I_{N_1} - A_{\Omega\Omega})^{-1} \Sigma_\Omega (I_{N_1} - A_{\Omega\Omega})^{-\top} \end{bmatrix} .$$

Then, if we denote $\widehat{Y} = \tau_0(X)$

$$cov((\widehat{Y}, \widehat{Z}_1)) = T cov(X) T^\top$$

and we can derive its conditional mean and covariance

$$\mu_{\widehat{Y}|\hat{z}_1} = \mu_{\widehat{Y}} + \bar{\tau}_0^\top (I_{N_0} - A_{00})^{-1} A_{0\Omega} (I_{N_1} - A_{\Omega\Omega})^{-1} \Sigma_{\pi(1)} (I_{N_1} - A_{\Omega\Omega})^{-\top} \bar{\tau}_1 \left( \bar{\tau}_1^\top (I_{N_1} - A_{\Omega\Omega})^{-1} \Sigma_\Omega (I_{N_1} - A_{\Omega\Omega})^{-\top} \bar{\tau}_1 \right)^{-1} (\hat{z}_1 - \mu_{Z_1}) .$$

Thus using the above equation

$$\mu_{\widehat{Y}|\hat{z}_1} = \mu_{\widehat{Y}} + \alpha (\hat{z}_1 - \mu_{Z_1}) = \bar{\tau}_0^\top (I_{N_0} - A_{00})^{-1} \mu_{\pi(0)} + \bar{\tau}_0^\top (I_{N_0} - A_{00})^{-1} A_{0\Omega} (I_{N_1} - A_{\Omega\Omega})^{-1} \mu_\Omega + \alpha \hat{z}_1 - \alpha \bar{\tau}_1^\top (I_{N_1} - A_{\Omega\Omega})^{-1} \mu_\Omega ,$$

which further simplifies with the same equation to

$$\mu_{\widehat{Y}|\hat{z}_1} = \mu_{\widehat{Y}} + \alpha (\hat{z}_1 - \mu_{Z_1}) = \bar{\tau}_0^\top (I_{N_0} - A_{00})^{-1} \mu_{U_0} + \alpha \hat{z}_1 .$$

Moreover,

$$\text{var}(\widehat{Y}|\hat{z}_1) = \sigma_{\widehat{Y}}^2 - \tau_0^\top (I_{N_0} - A_{00})^{-1} A_{0\Omega} (I_{N_1} - A_{\Omega\Omega})^{-1} \Sigma_\Omega (I_{N_1} - A_{\Omega\Omega})^{-\top} \bar{\tau}_1 \left( \bar{\tau}_1^\top (I_{N_1} - A_{\Omega\Omega})^{-1} \Sigma_\Omega (I_{N_1} - A_{\Omega\Omega})^{-\top} \bar{\tau}_1 \right)^{-1}$$

$$\bar{\tau}_1^\top (I_{N_1} - A_{\Omega\Omega})^{-1} \Sigma_\Omega (I_{N_1} - A_{\Omega\Omega})^{-\top} A_{0\Omega}^\top (I_{N_0} - A_{00})^{-\top} \bar{\tau}_0$$

again, using the above equation this leads to the simplification

$$\mathrm{var}(\widehat{Y}|\hat{z}_1) = \sigma_{\widehat{Y}}^2 - \alpha\bar{\tau}_1^\top(I_{N_1} - A_{\Omega\Omega})^{-1}\Sigma_{\pi(1)}(I_{N_1} - A_{\Omega\Omega})^{-\top}A_{0\Omega}^\top(I_{N_0} - A_{00})^{-\top}\bar{\tau}_0$$

$$= \sigma_{\widehat{Y}}^2 - \alpha\bar{\tau}_1^\top(I_{N_1} - A_{\Omega\Omega})^{-1}\Sigma_{\pi(1)}(I_{N_1} - A_{\Omega\Omega})^{-\top}\bar{\tau}_1\alpha$$

$$= \bar{\tau}_0^\top(I_{N_0} - A_{00})^{-1}\Sigma_{\pi(0)}(I_{N_0} - A_{00})^{-\top}\bar{\tau}_0\,.$$

For the high-level distribution we get

$$P(Y|z) = \mathcal{N}(\alpha z + \beta, \sigma_{Y|Z}^2)$$

where we can identify all parameters, including the mean and variance of the Gaussian exogenous variable $R_0$, with the above Eqs (22,24,25).

**Part 2: $n$-causes case $(n > 1)$**

We now assume $(\tau_k', \omega_k', \pi')_{k=1..n}$ such that the loss vanishes, but no such solution for $n + 1$ causes.

**Properties of exact $n$-cause solutions** Then such a solution can be linked to the 1-cause solution, which is guaranteed to exist according to our set of assumptions. Indeed, the existence of the n-cause solution implies that the pushfoward interventional distribution of the low-level causal model by $\tau$, $\widehat{P}$ satisfies

$$\widehat{P}^{(i)}(Y|\mathbf{Z} = z) \sim \mathcal{N}\left(\sum_k \alpha_k z_k + \beta, \sigma_{Y|Z}^2\right)$$

and

$$\widehat{P}^{(\omega(i))}(\mathbf{Z}) \sim \prod_k \widehat{P}^{(0)}(Z_k - \omega_k(i_k))$$

If we define the aggregate cause $\tilde{Z} = \sum_k \alpha_k z_k = \sum_k \alpha_k \tau_k(x)$, then we can rewrite the above model as

$$\widehat{P}^{(i)}(Y|\tilde{Z} = \tilde{z}) \sim \mathcal{N}\left(\tilde{z} + \beta, \sigma_{Y|Z}^2\right)$$

and

$$\widehat{P}^{(\omega(i))}(\tilde{Z}) \sim \prod_k \widehat{P}^{(0)}(\tilde{Z} - \sum_k \omega_k(i_k))$$

which implies that concatenating the $\tau_k$ with multiplicative coefficient $\alpha_k$ leads to a valid 1-cause TCR, and must thus match the expressions we have found for it, up to a multiplicative constant.

Moreover, the interventional consistency of the $n$-causes, which do not influence each other according to the assumed high-level causal graph, entails that any low-level intervention $i$ affects only high-level variable $Z_k$ through is components in $\pi(k)$.

We define $A_{1..n,1..n}$ by reordering the indices of $\Omega$ according to the assumed alignment $\pi$. Consistency then implies (using the above 1 cause solution proof)

$$\begin{bmatrix} \bar{\tau}_1^\top, & \dots, & 0 \\ \vdots, & \ddots, & \vdots \\ 0, & \dots, & \bar{\tau}_n^\top \end{bmatrix}(I_{\sum_k N_k} - A_{1..n,1..n})^{-1}i_\Omega = \begin{bmatrix} \omega_1^\top i_{\pi_1} \\ \vdots \\ \omega_n^\top i_{\pi_n} \end{bmatrix}. \tag{26}$$

This implies that

$$\bar{\tau}_k^\top(I_{\sum_k N_k} - A_{1..n,1..n})_{kj}^{-1} = 0 \text{ for all } j \in \pi(l), l \neq k\,, \tag{27}$$

where $(.)_{kj}$ indicates the matrix block corresponding to indices in $\pi(k) \times \pi(j)$. Because the non-vanishing coefficients of $\bar{\tau}_k$ reflect the influences along the causal pathways from nodes of $\pi(k)$ to $Y$, the above entails that $(I_{\sum_k N_k} - A_{1..n,1..n})_{kj}^{-1}$ must vanish on the support of $\bar{\tau}_k$. Indeed, otherwise Eq. (27) would indicate that causal pathways from nodes in $\pi(k)$ to $\pi(0)$ cancel each other, which is forbidden by our assumptions.

We thus deduce that any off-diagonal block element of $(I_{\sum_k N_k} - A_{1..n,1..n})^{-1}$ whose row component belongs to the support of any $\tau_k$ and whose column component belongs to the support of any $\omega_k$ must vanish. Indeed, otherwise the causes would

influence each other. In essence, this means that any node influencing the target must not influence any node though some causal pathway in another group with the same property.

**Identifiability of the $n$-cause solution** We assume $n = n_{max}$ and consider a second $n$-cause solution, which we denote: $(\tau'_k, \omega'_k, \pi')_{k=1..n}$.

*If* $\pi' = \pi$ up to a permutation of the order of the causes and removal of low-level variables that do not belong to the support of neither any $\omega$. Then identifiability of the corresponding 1-cause solution implies that each $\tau'_k$ is identified with each $\tau_k$ up to a multiplicative constant, because they both match the components of the 1-cause $\tau$ on their (identical) support $\pi(k)$ The same goes for $\omega'_k$ and $\omega_k$. This corresponds to the conclusion of the Proposition.

*Otherwise*, $\pi' \neq \pi$ even up to a permutation of the order of the causes and removal of low-level variables that do not belong to the support of neither any $\tau$ nor $\omega$. There should be an overlap between supports of omegas and taus of one cause of one solution with two different causes of the other solution. Without loss of generality, because of the block-diagonal structure of $(I_{\sum_k N_k} - A_{1..n,1..n})^{-1}$ entailed by Eqs. (26-27) for both solutions, this overlap implies that $\pi(k)$ for at least one $k$ can be further partitioned and reordered into two subgroups, such that the corresponding diagonal block of $(I_{\sum_k N_k} - A_{1..n,1..n})^{-1}$ can be turned in a block diagonal submatrix. This can be used to build a new alignment $\pi''$ for $n+1$ causes, and its associated tau and omega maps such that Eq. (26) will be again satisfied, leading to interventional consistency of the (n+1)-causes. Moreover, because the exogenous variables in $\Omega$ are assumed independent, the block diagonal structure of the newly defined matrix $(I_{\sum_k N_k} - A_{1..n+1,1..n+1})^{-1}$ entails that the covariance of the $n+1$ high-level cause variable will be diagonal, ensuring mutual independence between high-level causes. This procedure exhibits the existence of an $(n+1)$-cause TCR, contradicting the original assumption that $n = n_{max}$. This case is thus excluded. $\qquad\qquad\square$

# D ADDITIONAL THEORY

## D.1 REPARAMETRIZATIONS OF REDUCTIONS

In order to study invariance properties of TCR, we define transformations compatible with a class of reductions. Let $\rho : \mathcal{Z}_1 \times \cdots \times \mathcal{Z}_n \to \mathcal{Z}_1 \times \cdots \times \mathcal{Z}_n$ be a continuous invertible transformation of the $n$-dimensional high-level cause vector. Then the transformation

$$\tilde{\rho} : \begin{bmatrix} Y \\ \mathbf{Z} \end{bmatrix} \mapsto \begin{bmatrix} Y \\ \rho(\mathbf{Z}) \end{bmatrix}$$

is also continuous invertible. Among this class of transformations, we define an invertible reparametrization of a TCR as follows.

**Definition D.1.** *An invertible reparametrization of a reduction for the class $\mathcal{T}$ of $\tau$-maps and the class $\{\mathcal{H}_\gamma\}_{\gamma \in \Gamma}$ satisfies the following properties.*

- *it is* compatible *with the class of $\tau$-maps as follows: for any map $\tau \in \mathcal{T}$, we have $\tilde{\rho} \circ \tau \in \mathcal{T}$,*
- *it is* compatible *with the high-level model class $\{\mathcal{H}_\gamma\}$ as follows: for any model parameter $\gamma$, the uninterved and intervened distributions $P_{\mathcal{H},\gamma}(Y, \mathbf{Z})$ are such that there exist a parameter $\gamma'$ and a map between high-level interventions $\psi : \mathcal{J} \to \mathcal{J}$ such that the joint distributions of the transformed variables $(Y, \rho(\mathbf{Z}))$ is compatible with uninterved and intervened distributions of $\mathcal{H}_{\gamma'}$, in the sense that*

$$\tilde{\rho}_\#[P^{(j)}_{\mathcal{H},\gamma}(Y, \mathbf{Z})] = P^{(\psi(j))}_{\mathcal{H},\gamma'}(Y, \mathbf{Z}) .$$

## D.2 THE CASE OF A SINGLE TARGET LOW-LEVEL VARIABLE

Whenever $\pi(0)$ is a singleton, $\tau_0$ is univariate and the target $Y$ essentially corresponds (up to trivial rescaling) to a single low-level variable. We elaborate on the interpretation of Proposition 3.4 in this context.

Let us set $\bar{\tau}_0 = 1$ and fix the target index such that $\pi(0) = \{N\}$ without loss of generality. Then the DAG constraints entail $A_{00} = 0$ and the structural equations take the form

$$\mathbf{X}_{\pi(1)} := A_{11}\mathbf{X}_{\pi(1)} + \mathbf{U}_{\pi(1)} + i, \quad U_k \sim \mathcal{N}(\mu_k, \sigma_k^2) \tag{28}$$

$$Y := \mathbf{a}_{01}^\top \mathbf{X}_{\pi(1)} + U_N \tag{29}$$

where $\boldsymbol{a}_{01}$ is a column vector of coefficients of the low-level mechanism linking the target $Y$ to its causes in $\pi(0)$. Then the unique linear 1D TCR, up to a multiplicative constant, making the consistency loss vanish is given by

$$\bar{\tau}_1 = \boldsymbol{a}_{01} \tag{30}$$

$$\text{and} \quad \bar{\omega}_1 = (I_{N-1} - A_{11})^{-\top}\bar{\tau}_1 = (I_{N-1} - A_{11})^{-\top}\boldsymbol{a}_{01}. \tag{31}$$

This solution is easily interpretable: $\bar{\tau}_1$ identifies the ground truth mechanism linking $X_{\pi(0)}$ to the target, while $\bar{\omega}_1$ traces the contribution of interventions on each endogenous variable to the target. Indeed, this contribution is given by the "reduced form" map between exogenous values and endogenous values (see proof of Proposition 3.4 for more insights)

$$\boldsymbol{i} \mapsto (I_{N-1} - A_{11})^{-1}\boldsymbol{i},$$

and by composing this mapping with mechanism $\boldsymbol{a}_{01}$ we get the (shift) influence of interventions on the target

$$\boldsymbol{i} \mapsto \boldsymbol{a}_{01}^\top (I_{N-1} - A_{11})^{-1}\boldsymbol{i} = \bar{\omega}_1^\top \boldsymbol{i}.$$

The mismatch between $\bar{\omega}_1$ and $\bar{\tau}_1$ is due to the internal causal structure of the submodel described by eq. (28). Indeed, if there are no causal links within this subsystem, $A_{11}$ is a zeros matrix and

$$\bar{\omega}_1 = (I_{N-1})^{-\top}\bar{\tau}_1 = \bar{\tau}_1 = \boldsymbol{a}_{01},$$

otherwise, the two maps will be different. The discrepancy between the vectors thus reflects the fact that the causal explanation links high-level endogenous variables and interventions on them by potentially complex low-level interactions that do not necessarily have a simple high-level interpretation. This justifies regularizing the consistency loss with an homogeneity loss in order to focus on explanations that exhibit congruent $\tau$ and $\omega$ maps.

## D.3 THE CASE OF LINEAR CHAIN SCMS

In the case of a chain SCM

$$X_1 \to \cdots \to X_{N-1} \to X_N = Y$$

the above linear setting gets the additional constraints (using a causal ordering of the variables) that the target's mechanism is sparse

$$\boldsymbol{a}_{01}^\top = [0, \ldots, 0, a_N]$$

and the structure matrix of $X_{\pi(1)}$ is subdiagonal

$$A_{11} = \begin{bmatrix} 0 & 0 & \ldots & 0 & 0 \\ a_2 & 0 & \ldots & 0 & 0 \\ 0 & a_3 & \ldots & 0 & 0 \\ 0 & 0 & \ldots & 0 & 0 \\ 0 & 0 & \ldots & a_{N-1} & 0 \end{bmatrix}$$

and as a consequence, the solution writes

$$\bar{\tau}_1 = [0, \ldots, 0, a_{N-1}]^\top \tag{32}$$

$$\text{and} \quad \bar{\omega}_1 = (I_{N-1} - A_{11})^{-\top}\bar{\tau}_1 = \begin{bmatrix} a_2.a_3.\ldots.a_{N-1} \\ \vdots \\ a_{N-2}a_{N-1} \\ a_{N-1} \end{bmatrix}. \tag{33}$$

This solution is in line with our experimental results:

- $\bar{\tau}_1$ has all its weight on the parent of the target.
- $\bar{\omega}_1$ has a non-sparse distribution over the chains, decaying in the upstream direction. This reflects that structure coefficients of $A_{11}$ are selected with absolute value inferior to one, such that the influence of ancestor nodes on the target decays with their distance to it on the graph.

Transposing the chain example to the case of Proposition D.2, we can take the case were the direct parent $X_{N-1}$ of the target is left uninterved. In such a case, $\bar{\tau}_1$ may put its weight on both $X_{N-1}$ and its direct parent $X_{N-2}$, Proposition 3.4 provides two example solutions for different choices of $\pi(1)$, including or excluding $X_{N-1}$. In the most extreme case of dissimilarity between $\tau_1$ and $\omega_1$, solution including $X_{N-1}$ in $\pi(1)$ puts all $\tau_1$'s weight on $X_{N-1}$, while $\omega_1$ has no weight on it (because it is uninterved). As a consequence, $\omega_1$ and $\tau_1$ are orthogonal and the associated homogeneity loss vanishes. In contrast, the unique solution excluding $X_{N-1}$ from $\pi(1)$ have a larger cosine similarity and will thus be preferred by the homogenity-regularized loss.

## D.4 LOSS OF IDENTIFIABILITY THROUGH UNINTERVENED VARIABLES

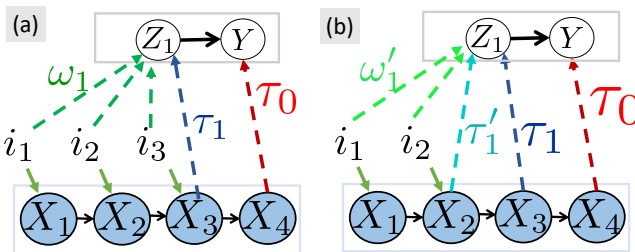

Figure 5: **1-cause TCR solutions on a chain graph.** Arrows indicate non-zero coefficients of each map. (a) Unique solution $\tau_1$ when interventions are performed on all nodes except the target. (b) Two solutions $\tau_1$ and $\tau_1'$ when only the first two nodes are interved on.

**Proposition D.2.** *Consider the setting of Prop. 3.4 with the exception that $\Omega \subsetneq \pi(1)$ such that there is now a non-empty subset $S = \pi(1) \setminus \Omega$, such that $X_\Omega \to X_S \to X_{\pi(0)}$. Then there exist at least two different linear 1D TCR such that $\mathcal{L}_{\mathrm{cons}} = 0$.*

This result can also be illustrated with a chain graph, as shown in Fig. 5(b). If the parent node $X_3$ of $Y = X_4$ is uninterved, then one may choose either $Z_1 = X_2$ or $Z_1 = X_3$ (matching the solution of Fig. 5(a)) to minimize $\mathcal{L}_{\mathrm{cons}}$. This is because both variables are equivalently mediating all performed interventions to $X_4$. Note that each choice has it own benefit: $Z_1 = X_3$, as a direct parent of $Y$, is a better statistical predictor of the value of $Y$. However, if we focus on causal interpretability of the high-level representation, $Z_1 = X_2$ is preferable because it is one of the variables interved on at the low-level as enforced by the prior $P(i)$, and such that it will be associated to a non-zero weight in $\omega_1$ for any solution satisfying $\mathcal{L}_{\mathrm{cons}} = 0$.

*Proof.* The low-level model follows the following SCM, with $P(i)$ non-trivial

$$X := AX + U + i, \quad U_k \sim \mathcal{N}(\mu_k, \sigma_k^2)$$

such that $X$, $A$ and $P$ take the block forms

$$X = \begin{bmatrix} X_{\pi(0)} \\ X_S \\ X_\Omega \end{bmatrix}, \quad A = \begin{bmatrix} A_{00} & A_{0S} & \mathbf{0} \\ \mathbf{0} & A_{SS} & A_{S\Omega} \\ \mathbf{0} & \mathbf{0} & A_{\Omega\Omega} \end{bmatrix},$$

with $\pi(0)$ of size $N_0$, and $\pi(1)$ of size $N_1 = N - N_0$ and $S$ of size $s$. Then we know from the Proposition 3.4 that there is already a valid solution using $\pi$ as alignment. The only difference is that variables in $S$ are uninterved, which does not affect the ability of the solution to achieve $\mathcal{L}_{\mathrm{cons}} = 0$. That solution would be compatible with interventions on $S$, but since $S$ is uninterved, we do not have uniqueness guarantees for this choice of $\pi$.

Alternatively, if we choose $\pi'(0) = \pi(0) \cup S$ and $\pi'(1) = \pi(1) \setminus S = \Omega$, then, we can again apply Propostion D.2, and see that it provides a different solution with this alignment, which is compatible with the given problem (constructive transformation with constraint on the mapping $\tau_0$). Importantly, the key indeterminacy is for the map $\tau_1$, which will either put all its weight on elements in $S$ (direct parents of $\pi(0)$), or alternatively, put all its weights on elements in $\Omega$. There is an additional, but trivial, indeterminacy for the map $\omega_1$: indeed, since $X_S$ is uninterved (as part of $\pi(0)$), the weights in $\omega_1$ associated to these coefficients may take arbitrary values (since their associated component in $i$ remains zero). We do not consider these trivial indeterminacies (which do not affect the mapping $\omega_1$ on its domain, i.e. the support of the prior $P(i)$) by forcing the weights of $\omega_1$ associated to uninterved variables to zero. $\square$

## D.5    CONNECTION TO CAUSAL ABSTRACTIONS

**Proposition D.3.** *Assume the low-level SCM follows*

$$X := AX + U + i, \quad U_k \sim \mathcal{N}(\mu_k, \sigma_k^2), \sigma_k^2 > 0, \; i \sim P(i)$$

*such that $X$ and $A$ take the block forms*

$$X = \begin{bmatrix} X_{\pi(0)} \\ X_{\pi(1)} \end{bmatrix}, \quad A = \begin{bmatrix} A_{00} & A_{01} \\ 0 & A_{11} \end{bmatrix}.$$

*Given an arbitrary choice of linear scalar target of the form $Y = \tau_0^\top X = \bar{\tau}_0^\top X_{\pi(0)}$, under the conditions of Proposition 3.4, the unique linear 1-cause TCR (up to a multiplicative constant) is associated to a 1-cause constructive abstraction given by*

$$\bar{\tau}_1 = A_{01}^\top (I_{\#\pi(0)} - A_{00})^{-\top} \bar{\tau}_0 \tag{34}$$

$$\bar{\omega}_1 = (I_{\#\pi(1)} - A_{11})^{-\top} \bar{\tau}_1, \tag{35}$$

$$\bar{\tau}_{U,0} = \bar{\tau}_0^\top (I_{N_0} - A_{00})^{-1}, \tag{36}$$

$$and \; \bar{\tau}_{U,1} = \bar{\tau}_1^\top (I_{N_1} -)^{-1}. \tag{37}$$

*Proof.* To have a valid constructive causal abstraction, we need to verify the existence of an additional constructive map $\tau_U$ for exogenous variables such that for all realizations $u$ of $U$.

$$\tau(\mathcal{L}^{(i)}(u)) = \mathcal{H}^{(\omega(i))}(\tau_U(u))$$

where $\mathcal{L}^{(i)}(.)$ and $\mathcal{H}^{(j)}(.)$ denote the mappings from endogenous to exogenous variable for the low- and high-level intervened models, respectively.

Using

$$\mathcal{L}^{(i)}(U) = X^{(i)} = (I_N - A)^{-1}(U + i)$$

we get

$$X^{(i)}_{\pi(1)} = (I_{N_1} -)^{-1}(U_{\pi(1)} + i_{\pi(1)})$$

and

$$X^{(i)}_{\pi(0)} = (I_{N_0} - A_{00})^{-1} U_{\pi(0)} + (I_{N_0} - A_{00})^{-1} A_{01} X^{(i)}_{\pi(1)}.$$

Applying the $\tau_0$ map we get the solution of Prop. 3.4

$$Y^{(i)} = \bar{\tau}_0^\top X^{(i)}_{\pi(0)} = \bar{\tau}_0^\top (I_{N_0} - A_{00})^{-1} U_{\pi(0)} + \bar{\tau}_0^\top (I_{N_0} - A_{00})^{-1} A_{01} X^{(i)}_{\pi(1)} = \bar{\tau}_0^\top (I_{N_0} - A_{00})^{-1} U_{\pi(0)} + \bar{\tau}_1^\top X^{(i)}_{\pi(1)}$$

moreover applying the $\tau_1$ map to the variables in $\pi(1)$ we get

$$\bar{\tau}_1^\top X^{(i)}_{\pi(1)} = \bar{\tau}_1^\top (I_{N_1} -)^{-1}(U_{\pi(1)} + i_{\pi(1)}) = \bar{\tau}_1^\top (I_{N_1} - A_{\Omega\Omega})^{-1} U_{\pi(1)} + \bar{\omega}_1^\top i_{\pi(1)}$$

So by defining the vectors $\bar{\tau}_{U,1}^\top = \bar{\tau}_1^\top (I_{N_1} -)^{-1}$ and $\bar{\tau}_{U,0}^\top = \bar{\tau}_0^\top (I_{N_0} - A_{00})^{-1}$ we get a valid constructive abstraction linking the low-level map $u \mapsto \mathcal{L}^{(i)}(u)$ to the following high-level map (first component is $Y$, second is the cause $Z_1$)

$$r \mapsto \mathcal{H}^{(j)}(r) = \begin{bmatrix} r_0 + r_1 + j_1 \\ r_1 + j_1 \end{bmatrix}$$

such that for all $u$

$$\mathcal{H}^{(\bar{\omega}_1^\top i_{\pi(1)})}\left( \begin{bmatrix} \bar{\tau}_{U,0}^\top u_{\pi(0)} \\ \bar{\tau}_{U,1}^\top u_{\pi(1)} \end{bmatrix} \right) = \tau(\mathcal{L}^{(i)}(u))$$

$\square$

.

# E ALGORITHM DETAILS

## E.1 GAUSSIAN CONSISTENCY LOSS

As the KL divergence is hard to estimate in the non-parametric setting, we make a Gaussian approximation of this loss to get an analytical, differentiable expression. Using the general formula for two n-dimensional Gaussian densities $P$ and $Q$

$$\text{KL}(P||Q) = \frac{1}{2}\left[(\mu_Q - \mu_P)^\top \Sigma_Q^{-1}(\mu_Q - \mu_P) + \text{tr}(\Sigma_Q^{-1}\Sigma_P) - \log\frac{|\Sigma_P|}{|\Sigma_Q|} - n\right].$$

Parameters of the reduction are $\tau_k, \mu_Z, \mu_{Y|Z}, f : z \to f(z), \omega_k$ with

$$Z^{(\omega(i))} \sim P(z) = \mathcal{N}(\mu_Z + Wi, \Sigma_Z), \text{ with } W = [\omega_1, ..., \omega_n]^\top \text{ and } \Sigma_Z = \text{diag}(\sigma_{Z,1}^2, ..., \sigma_{Z,n}^2))$$
$$Y^{(\omega(i))}|z \sim P(Y|z) = \mathcal{N}(f(z), \sigma_{Y|Z}^2),$$
$$\hat{Z}^{(i)} = [\tau_1, ..., \tau_n]^\top X^{(i)} = TX^{(i)},$$
$$\hat{Y}^{(i)} = \tau_0^\top X^{(i)}.$$

Moreover, we estimate the second order properties of the simulator distribution for each intervention $i$

$$\hat{\mu}_X^{(i)} = \langle X^{(i)}\rangle,$$
$$\hat{\Sigma}_X^{(i)} = \left\langle \left(X^{(i)} - \hat{\mu}_X^{(i)}\right)^\top \left(X^{(i)} - \hat{\mu}_X^{(i)}\right)\right\rangle,$$
$$\hat{\mu}_Z^{(i)} = \langle\hat{Z}^{(i)}\rangle = T\hat{\mu}_X^{(i)},$$
$$\hat{\mu}_Y^{(i)} = \langle\hat{Y}^{(i)}\rangle = \tau_0^\top \hat{\mu}_X^{(i)},$$
$$\widehat{\Sigma}_Z^{(i)} = \left\langle\left(\widehat{Z}^{(i)} - \hat{\mu}_Z^{(i)}\right)\left(\widehat{Z}^{(i)} - \hat{\mu}_Z^{(i)}\right)^\top\right\rangle = T\hat{\Sigma}_X^{(i)}T^\top,$$
$$\widehat{\sigma^2}_{Z,k}^{(i)} = \left(\widehat{\Sigma}_Z^{(i)}\right)_{k,k} = \left\langle\left(\widehat{Z}_k^{(i)} - \hat{\mu}_{Z,k}^{(i)}\right)^2\right\rangle = \tau_k^\top \hat{\Sigma}_X^{(i)}\tau_k,$$
$$\widehat{\sigma^2}_Y^{(i)} = \left\langle\left(\hat{Y}^{(i)} - \hat{\mu}_Y^{(i)}\right)^2\right\rangle = \tau_0^\top\hat{\Sigma}_X^{(i)}\tau_0,$$
$$\widehat{c}_{ZY}^{(i)} = \left\langle\left(\hat{Y}^{(i)} - \hat{\mu}_Y^{(i)}\right)\left(\hat{Z}^{(i)} - \hat{\mu}_Z^{(i)}\right)\right\rangle = T\hat{\Sigma}_X^{(i)}\tau_0,$$

where $\langle\cdot\rangle$ denotes the empirical average. Using the KL between Gaussian variables, we can rewrite the consistency loss as

$$\mathcal{L}_{\text{cons}} = \mathbb{E}_{i\sim p(i)}\left[\text{KL}_z(\hat{P}^{(i)}(z)|P^{(\hat{\omega}(i))}(z)) + \mathbb{E}_{z\sim\hat{P}^{(i)}(Z)}\left[\text{KL}_Y\left(\hat{P}^{(i)}(Y|Z=z)||P^{(0)}(Y|Z=z)\right)\right]\right]$$

$$= \frac{1}{2}\mathbb{E}_{i\sim p(i)}\left[\sum_k\left(\frac{(\mu_{Z,k} + \omega_k^\top i - \hat{\mu}_{Z,k}^{(i)})^2}{\sigma_{Z,k}^2} + \frac{\widehat{\sigma^2}_{Z,k}^{(i)}}{\sigma_{Z,k}^2}\right) - \ln\left(\frac{|\widehat{\Sigma}_Z^{(i)}|}{\prod_k \sigma_{Z,k}^2}\right) - n\right]$$

$$+ \frac{1}{2}\mathbb{E}_{i\sim p(i),z\sim\hat{P}^{(i)}(Z)}\left[\frac{\left(f(z) - \hat{\mu}_Y^{(i)} - \left(\widehat{c}_{ZY}^{(i)}\right)^\top\left(\widehat{\Sigma}_Z^{(i)}\right)^{-1}(z - \hat{\mu}_Z^{(i)})\right)^2}{\sigma_{Y|Z}^2}\right.$$

$$\left. + \frac{\widehat{\sigma^2}_Y^{(i)} - \left(\widehat{c}_{ZY}^{(i)}\right)^\top\left(\widehat{\Sigma}_Z^{(i)}\right)^{-1}\widehat{c}_{ZY}^{(i)}}{\sigma_{Y|Z}^2} - \ln\left(\frac{\widehat{\sigma^2}_Y^{(i)} - \left(\widehat{c}_{ZY}^{(i)}\right)^\top\left(\widehat{\Sigma}_Z^{(i)}\right)^{-1}\widehat{c}_{ZY}^{(i)}}{\sigma_{Y|Z}^2}\right) - 1\right]. \quad (38)$$

## E.2 ADDITIONAL INFORMATION

The overall algorithm is described in the Algorithm 1 of main text, and it implementation can be found in the file `targeted_causal_reduction/model/causal_pattern_reduction.py`.

# F EXPERIMENTAL DETAILS

## F.1 LINEAR EXPERIMENTS

| Parameters | Linear (Fig. 2a) | Two Branch (Fig. 2b) |
|---|---|---|
| learning rate $\lambda$ | $10^{-3}$ | $10^{-3}$ |
| learning rate scheduler | - | cosine annealing |
| No. repeated train. runs per seed | 1 | 10 |
| simulation paths $n_{\text{sim}}$ | 10, 000 | 10, 000 |
| training epochs $N_{\text{ite}}$ | 100 | 600 |
| simulation batch size $B$ | 128 | 128 |
| intervention batch size $B_i$ | 64 | 512 |
| overlap reg. $\eta_{\text{ov}}$ (12) | 0 | 0.1 |
| balancing reg. $\eta_{\text{bal}}$ (13) | 0 | $10^{-3}$ |

Table 1: **Experimental parameters and settings for the linear Gaussian experiments.**

**Sampling linear Gaussian low-level models** For the adjacency matrix, we sample all non-zero entries uniformly in the interval $[-1, 1]$. For general adjacency matrices, the lower triangular elements of the adjacency matrix are non-zero, where we assume that the target $Y$ has only incoming edges and the variables are arranged in topological order. For the two-branch graph, values in the adjacency are set to zero accordingly. For chain graphs, the first lower off-diagonal entries are non-zero. The exogenous variables $U$ and shift interventions $i$ are independent Gaussian with $U_j, i_j \sim \mathcal{N}(0, 1)$ for $j = 1, ..., N$.

**Data and Training** The data and training parameters are summarized in Table 1. All simulation data is generated before training and reused in each epoch. We split the data into training (70%), validation and test (15% each). Since the training of the two-variable model would occasionally get stuck in local minima, we run each training with 10 different random initializations of the weights and select the model with the best total validation loss (14) at the end of training. Furthermore, we use a cosine annealing learning rate scheduler with a final learning rate of $10^{-5}$.

## F.2 DOUBLE WELL

**Simulation** We model the ball moving in a double well potential $V(x) = x^4 - 4x^2$, shown in Figure 3(a), by the following equation of motion:

$$m\ddot{x}(t) + k\dot{x}(t) + \frac{\partial}{\partial x}V(x(t)) = 0 \quad \Rightarrow \quad m\ddot{x}(t) + k\dot{x}(t) + 4x(t)^3 - 8x(t) = 0, \tag{39}$$

where $x(t)$ is the position of the ball at time $t$, $\dot{x}(t)$ and $\ddot{x}(t)$ are the first and second time derivatives, respectively, $k$ is the friction coefficient and $m$ is the mass of the ball. We can reformulate the second order ODE into a system of first order ODEs by introducing the velocity $v(t) = \dot{x}(t)$ as a variable:

$$\dot{x}(t) = v(t)$$
$$\dot{v}(t) = -\frac{1}{m}\left(kv(t) + 4x(t)^3 - 8x(t)\right). \tag{40}$$

We solve the system of ODEs numerically on a grid of 101 time points $t_k$ for $k = 0, \ldots, 100$ equally spaced between $t = 0$ and $t = 10$ using a numerical integration method. The initial conditions are $x(0) = -2.07414285 + 5 \times 10^{-7} \times \varepsilon_x$, with $\varepsilon_x \sim \text{Uniform}(-1, 1)$ and $v(0) = 11$. The initial values are chosen such that there is a non-zero chance that the ball ends up in the left or right well without any additional interventions.

For shift interventions, we sample random velocity shifts $\Delta v(t_k) \sim \mathcal{N}(0, 0.5)$. The positions are unshifted. In the numerical integration scheme, the shift interventions are implemented by splitting the integration domain in parts. The ODE system is integrated from the initial conditions at $t_0$ to the next time grid at $t_1$. Then the velocity at $t_1$ is shifted by $\Delta v(t_1)$ and used as the initial value for the next integration starting at $t_1$, and so on. Similarly, we introduce independent stochasticity by adding noise to the velocity sampled from $\mathcal{N}(0, 0.2)$ at each time step, mimicking intrinsic noise of the system.

| Parameters | Double Well (Fig. 3) |
|---|---|
| learning rate $\lambda$ | $5 \cdot 10^{-4}$ |
| learning rate scheduler | - |
| No. repeated train. runs per seed | 1 |
| simulation paths $n_{\text{sim}}$ | $10,000$ |
| training epochs $N_{\text{ite}}$ | 200 |
| simulation batch size $B$ | 128 |
| intervention batch size $B_i$ | 64 |
| overlap reg. $\eta_{\text{ov}}$ (12) | 0 |
| balancing reg. $\eta_{\text{bal}}$ (13) | 0 |

Table 2: **Experimental parameters and settings for the double well experiments.**

**Data and Training**    The data and training parameters are summarized in Table 1. All simulation data is generated before training and reused in each epoch. We split the data into training (70%), validation and test (15% each).

## F.3    SPRING-MASS SYSTEM

| Parameters | 4 masses with different weights (Fig. 4) | 2 groups of masses (Fig. 7) |
|---|---|---|
| learning rate $\lambda$ | $10^{-4}$ | $10^{-3}$ |
| learning rate scheduler | cosine annealing | cosine annealing |
| No. repeated train. runs per seed | 5 | 5 |
| simulation paths $n_{\text{sim}}$ | $10,000$ | $10,000$ |
| training epochs $N_{\text{ite}}$ | $4,800$ | $1,800$ |
| simulation batch size $B$ | 128 | 128 |
| intervention batch size $B_i$ | 64 | 512 |
| overlap reg. $\eta_{\text{ov}}$ (12) | 0.1 | 0.1 |
| balancing reg. $\eta_{\text{bal}}$ (13) | 0.1 | 0.1 |
| spring constant $k$ | $10^{-3}$ | $10^{-3}$ |
| rest length $u_0$ | 1 | 1 |
| masses $m_i$ | $(0.5, 0.83, 0.17, 1.5)$ | all 1 |

Table 3: **Experimental parameters and settings for the spring mass system experiments.**

**Simulation**    Let $M$ be the number of masses. Then, $m_i \in \mathbb{R}$, $\tilde{\vec{x}}_i(t) \in \mathbb{R}^2$ and $\vec{v}_i(t) \in \mathbb{R}^2$ represent the weight, position and velocity of mass $i = 1, \ldots, M$ at time $t$. $A \in \{0, 1\}^{M \times M}$ is the adjacency matrix encoding the spring connections, where $A_{ij} = 1$ indicates that a spring connects masses $i$ and $j$. The rest length at which the springs exert no force is denoted by $u_0$ and $k$ is the spring constant. Both $u_0$ and $k$ are assumed to be the same for all springs.

The total force acting on mass $i$ at time $t$ is given by

$$\vec{F}_i(t) = -k \sum_{j, A_{ij}=1} \left( \|\vec{u}_{ij}(t)\| - u_0 \right) \frac{\vec{u}_{ij}(t)}{\|\vec{u}_{ij}(t)\|} \tag{41}$$

where $\vec{u}_{ij}(t) = \vec{x}_i(t) - \vec{x}_j(t)$ is the displacement vector from mass $j$ to mass $i$. The equations of motion are

$$\frac{\mathrm{d}\tilde{\vec{x}}_i(t)}{\mathrm{d}t} = \vec{v}_i(t), \qquad \frac{\mathrm{d}\vec{v}_i(t)}{\mathrm{d}t} = \vec{a}_i(t), \quad \text{with} \quad \vec{a}_i(t) = \frac{\vec{F}_i(t)}{m_i}. \tag{42}$$

We assume that the masses have no volume and do not collide or interact other than the forces coming from the springs.

We solve the system of ODEs numerically on a grid of 21 time points $t_k$ for $k = 0, \ldots, 20$ equally spaced between $t = 0$ and $t = 100$ using a numerical integration method. The positions are initially set on a grid to $\vec{\tilde{x}}_1(t = 0) = (0, 0) + \vec{\tilde{x}}_{\text{offset}}$, $\vec{\tilde{x}}_2(t = 0) = (1, 0) + \vec{\tilde{x}}_{\text{offset}}$, $\vec{\tilde{x}}_3(t = 0) = (0, 1) + \vec{\tilde{x}}_{\text{offset}}$ and $\vec{\tilde{x}}_4(t = 0) = (1, 1) + \vec{\tilde{x}}_{\text{offset}}$, where $\vec{\tilde{x}}_{\text{offset}} \sim \mathcal{N}(0, 10)$ is a random offset that shifts the entire system. The initial velocities are independently drawn as $\vec{v}_i(t = 0) \sim \mathcal{N}(0, 0.01)$. We apply random independent velocity shifts $\Delta\vec{v}_i(t_k) \sim \mathcal{N}(0, 0.005)$ at each time step and integrate it into the ODE solver in the same way as for the double well experiment in App. F.2.

The feature vectors $X$ used to learn the TCR of the spring-mass system consists of all velocity values for all masses across all simulated time points. The interventions $i$ are the corresponding velocity interventions.

**Data and Training**     The data and training parameters are summarized in Table 3. All simulation data is generated before training and reused in each epoch. We split the data into training (70%), validation and test (15% each). Similar to the experiments on the two-branch linear graph in App. F.1, we repeat the training runs with different weight initializations and use a cosine annealing learning rate scheduler.

## F.4    ADDITIONAL RESULTS

### F.4.1    Spring-Mass System without Regularization

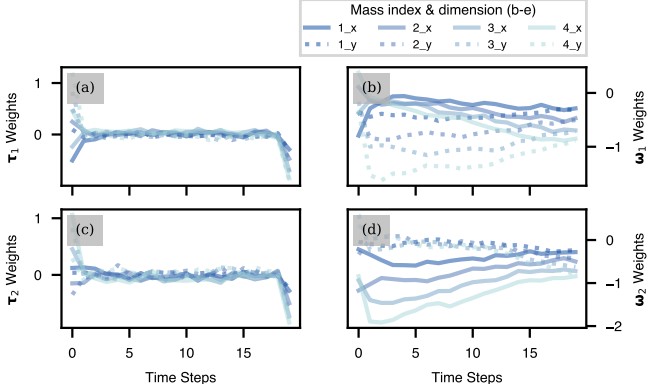

Figure 6: **Spring-mass system experiment without regularization.** Same experimental setup as described in Sec. 5.3 and App. F.3 with the regularization turned off, *i.e.* $\eta_{\text{ov}} = \eta_{\text{bal}} = 0$. The learned high-level mechanism is $f(\mathbf{Z}) \approx -0.180Z_1 + 0.125Z_2$.

When running the TCR algorithm without regularization, it cannot be ensured that the found solutions correspond to different properties of the low-level system, as shown in Fig. 6. There is significant mixing among the high-level variables, in particular the velocity in $x$-direction of the masses towards the end of the simulation appears in both high-level variables.

### F.4.2    Grouped Spring-Mass System

We simulate two groups of four masses as shown in Fig. 7(a). In contrast to the experiment shown in Sec. 5.3, all masses have equal weight and the target is the center of mass velocity in $x$-direction at the end of the simulation. The data and training parameters are summarized in Table 3.

Since the only interactions between masses are mediated by the springs, as described in App. F.3, the two groups of masses do not influence each other and are thus fully independent. The learned TCR identifies the two groups of masses as the two independent causes of the target. This is reflected in the parameters shown in Fig. 7 (b-e), where high-level variable $Z_1$ is predominantly influenced by the behavior of the second group (yellow) and variable $Z_2$ by the first group (blue). Furthermore, we observe that the $y$-component of the velocity, which is irrelevant for the target here, is ignored by the TCR and filtered out.

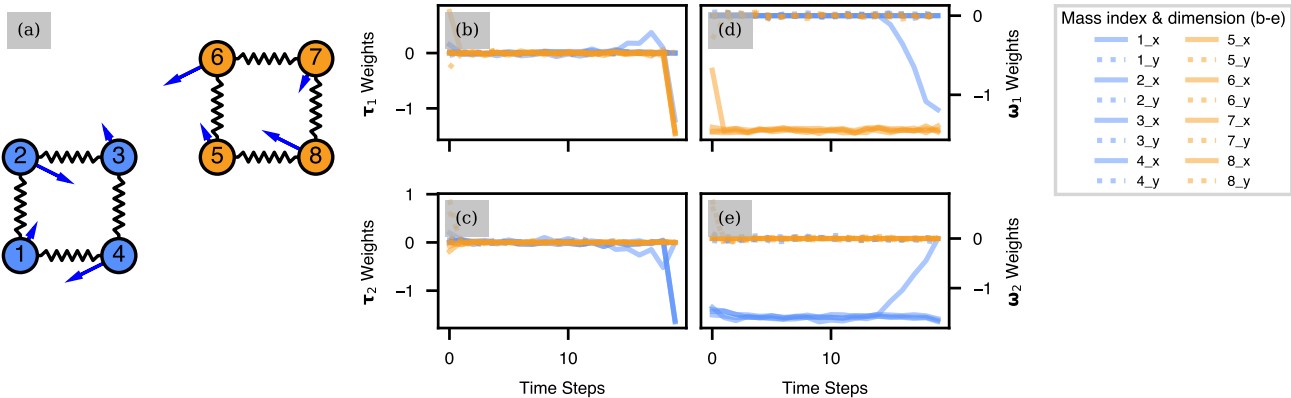

Figure 7: **Spring-mass system experiment with two groups of masses.** (a) Simulated system of eight point masses with equal weights connected by springs in two groups of 4 and with random initial velocity (blue arrows). In contrast to the experiment shown in Sec. 5.3, target of the simulation is the center of mass speed in $x$-direction. (b-e) Learned $\tau$- and $\omega$-weights corresponding to velocity components in $x$- and $y$-direction for a TCR with two high-level variables. The learned high-level mechanism is $f(\mathbf{Z}) \approx -0.0866 Z_1 - 0.0782 Z_2$.