# OpenReview forum: "Targeted Reduction of Causal Models"
_auai.org/UAI/2024/Conference — UAI 2024 oral_

### Official Review · Reviewer_Ufeg · 2024-03-18

**Q2-1 Originality-Novelty:** 3
**Q2-2 Correctness-Technical Quality:** 3
**Q2-5 Clarity Of Writing:** 3

**Q1 Summary And Contributions:**

The authors propose a method called Targeted Causal Reduction (TCR), which simplifies complex causal representation learning problems into learning a compact set of causal factors for a specific target phenomenon.

**Q2-3 Extent To Which Claims Are Supported By Evidence:**

3: Good: the main claims are supported by convincing evidence (in the form of adequate experimental evaluation, proofs, (pseudo-)code, references, assumptions).

**Q2-4 Reproducibility:**

2: Fair: key resources (e.g. proofs, code, data) are unavailable but key details (e.g. proof sketches, experimental setup) are sufficiently well-described for an expert to confidently reproduce the main results.

**Q3 Main Strengths:**

1. The proposed method has some novelties to causal representation learning.
2. The authors conducted experiments on both synthetic models and scientific simulations.

**Q4 Main Weakness:**

1. The assumptions including exact transformation, Gaussian approximations and etc. are strong.
2. No major weaknesses found. See comments for detailed questions.

**Q5 Detailed Comments To The Authors:**

1. In section 2.1, third paragraph, why does the probability of masking interventions grow as the number of low-level variables grows?
2. In Def 2.3, $k \neq l$ should be $k\neq n$.
3. The notation paragraph in section 3.3 could be put in previous parts so that it is easier for readers to understand.
4. In the definition of Omega map inspection 3.3, what is the value of vector $i$?
5. Is it true that $\pi(0)$ of the low-level variables are assumed to be the variables that are not interminable? And thus #$\Omega+|\pi(0)|=N$?
6. How are the hyperparameters like $\eta_{ov}, \eta_{bal}$ tuned for each experiment? In practice, how hard would this be?

**Q9 Complying With Reviewing Instructions:**

Yes

---

> ### Author Rebuttal · Authors · 2024-04-04
>
> Thank you for taking the time to provide detailed feedback.
>
> ### **Reply to Q4**
>
> We agree that the Gaussian assumption is rather strong, but was necessary to make the consistency loss tractable. We elaborate a bit more on this and other assumptions made in the response to Reviewer 2hcr. However, exact transformations are not required by the approach, which searches for transformations that are “as exact as possible”.
>
> ### **Answers to detailed comments**
>
> 1. *Probability of masking interventions*
>
> Here, we tried to motivate why do-interventions are a poor choice for our problem and we use soft interventions instead. Do-interventions set the value of a variable to some value and remove the influence of its parent variables. Hence, no information of interventions applied to upstream variables (parents, parents of parents and so on) is propagated to the target. This is what we refer to as masking in this setting. Now, if we sample  interventions on each variable independently from each other and from the same distribution, one would expect the probability of some interventions being (partially) masked by others to increase with a growing number of variables in the generic case (unless the low-level has a very specific causal graph such as a star shape). Strategies to avoid such masking could be (i) to apply do-intervention on a single low-level variable at a time, or (ii) exploit additional knowledge of the low-level causal relationships to choose which variables can be intervened on simultaneously.
>
> Soft interventions do not have these limitations and we can intervene on multiple (or all) variables simultaneously without knowing the low-level causal structure.
>
> 2. and 3.
>
> Thank you for pointing this out. We will correct this in the camera-ready version.
>
> 4. *“value of vector $\mathbf{i}$”*
>
> The intervention vector $\mathbf{i}$ is sampled from a prior intervention distribution $P(\mathbf{i})$ (see section 3.1), and in practice have jointly independent Gaussian densities in our experiments. Interventions can be thought of as probing or disturbing the system and are used as a learning signal for the reduction. The intervention distribution should reflect reasonable changes in the system which have to be determined through domain knowledge. For example, interventions correspond to shifts in velocity of the ball in the double well experiment (Section 5.2).
>
> 5. *Set of non-intervenable variables*
>
> Yes, in the setting of Proposition 3.4, $\pi(0)$ are all the unintervened low-level variables, and the target can potentially (but not necessarily) depend on them, such that $Y=\tau_0(\mathbf{X}_{\pi(0)})$. This is however not a requirement of the approach outside the scope of this theoretical result. Since we look for a constructive transformation and use a high-level model that does not allow direct intervention on the target variable $Y$ (to explain changes in Y by the influence of clearly distinct high-level causes), interventions on variables in $\pi(0)$ are not accounted for in the high-level model and may lead to transformations that are not exact, which is allowed by our framework.
>
> 6. *Hyperparameter tuning*
>
> In our experiments, we start by training the reduction without overlap and balancing loss. This gives a good baseline for the achievable consistency loss. Then, the regularization parameters are increased in further training runs while checking that the consistency loss does not deteriorate. We then choose regularization hyperparameters which have low regularization penalty while not sacrificing consistency.
>
> Assessing the difficulty is somewhat subjective, but from our experience with other model types it was very manageable. The tuning fails in ways that you would expect: when one or both regularization parameters are too high, the reduction does not achieve good consistency (relative to the solution without regularization). Thus, consistency is sacrificed in order to minimize the regularization penalty. When training with insufficient regularization, the reduction collapses to a solution with only one high-level cause, while the others are ignored (vanishing weights in the high-level mechanism). This is reflected in a poor validation loss on the balancing loss. That is the training behaviour one can check for when tuning. Furthermore, it was enough to sweep the regularization parameters on a logarithmic scale; i.e. the results did not critically depend on minor differences in the hyperparameters. Another difficulty that we have observed is that the reduction can get stuck in local minima. We solve this by repeatedly training with different initialisations of the model weights and picking the reduction according to the best total loss in validation.

---

### Official Review · Reviewer_JC7R · 2024-03-20

**Q2-1 Originality-Novelty:** 4
**Q2-2 Correctness-Technical Quality:** 3
**Q2-5 Clarity Of Writing:** 3

**Q1 Summary And Contributions:**

This is a very interesting paper and a great first step towards an interesting line of research. The theory is restricted to shift interventions and linear models, and the algorithm seems to require substantial tuning in practice, but I like the overall idea and the first step solution.

**Q2-3 Extent To Which Claims Are Supported By Evidence:**

3: Good: the main claims are supported by convincing evidence (in the form of adequate experimental evaluation, proofs, (pseudo-)code, references, assumptions).

**Q2-4 Reproducibility:**

3: Good: key resources (e.g. proofs, code, data) are available and key details (e.g. proofs, experimental setup) are sufficiently well-described for competent researchers to confidently reproduce the main results.

**Q3 Main Strengths:**

Novelty, clarity of writing, discussion, proofs

**Q4 Main Weakness:**

Questions about balancing loss, n_max experiments and no scale interventions per below

**Q5 Detailed Comments To The Authors:**

pg 2: would include reasoning behind why you consider only mutually independent high level variables Z

Section 3.4: would include discussion about scaling interventions, since you are focusing on linear models. I understand the theory for shift interventions, but I do not know why you do not consider scale interventions. Where does the theory break down? Readers will naturally wonder about scale interventions when they read "linear models."

Proposition 3.4: the first conclusion of this proposition, that there is always a one-cause model, is somewhat unsatisfying because it is effectively states that we cannot disentangle a high level model with multiple causes in the linear case. The second conclusion with n_max is more interesting, but none of the experiments use n_max, even in the two branch model of Section 5.1. Instead, you introduce a balancing loss, which was unexpected.

Why is the balancing loss needed if the n_max cause TCR is unique (up to rescaling and permutation)? Is this a practical convergence issue?

**Q9 Complying With Reviewing Instructions:**

Yes

---

> ### Author Rebuttal · Authors · 2024-04-04
>
> We thank you for the insightful observations and the investment of your time in reviewing our paper.
>
> ### **Answers to the detailed comments**
>
> *Constraint of mutually independent high-level causes*
>
> Our main goal in this work is to provide simple and interpretable high-level explanations of target phenomena. To ensure straightforward interpretations, we chose the simplest high-level causal structure. We give a more detailed answer in the response to Reviewer 2hcr.
>
> *Scaling interventions*
>
> Shift interventions can be combined with linear models and linear reduction maps to turn a low-level shift intervention into a high-level shift intervention. This allows satisfying exact transformation relations within this model class.  Consider a vector $\boldsymbol{X}$ of a subset of a low-level model with $N$ variables, linked by a matrix $A$ of low-level causal coefficients and the associated exogenous vector $\boldsymbol{U}$. Using vector $\boldsymbol{\tau}$ to map them to a high-level variable $Z$, we have for low-level shift intervention $\boldsymbol{i}$:
> $$Z=\boldsymbol{\tau}^\top (I_N-A)^{-1} (\boldsymbol{U}+\boldsymbol{i})=\boldsymbol{\tau}^\top (I_N-A)^{-1} \boldsymbol{U}+\boldsymbol{\tau}^\top (I_N-A)^{-1}\boldsymbol{i}$$
> showing the image of the intervened $X$ by ${\tau}$ can be decomposed as the sum of the image of the unintervened $X$ by $\boldsymbol{\tau}$ and a high-level shift intervention.
>
> For scaling interventions, we can assume the intervened $Z$ would for instance take the form $$Z=\boldsymbol{\tau}^\top (I_N-\text{diag}(\boldsymbol{i})A)^{-1} \boldsymbol{U}\,,$$ with $\text{diag}(\boldsymbol{i})$ the diagonal matrix gathering the interventional scaling coefficients. In the generic case, we do not see a clear decomposition of this expression in the form of a standard family of high-level interventions. Hence, we left extensions to other types of interventions to future work. We will comment on this point as requested.
>
> *Disentanglement of multiple causes in Proposition 3.4*
>
> The one-cause solution exhibited in Proposition 3.4 indeed does not address disentanglement of multiple high-level causes when they exist. Here, we wanted to show that there is a solution with perfect consistency under additional simplifying assumptions. The statement with the maximal number of independent causes addresses this disentanglement by showing the uniqueness of the solution. However, how to learn this solution is left open by this proposition. Running the TCR algorithm with multiple high-level causes without balancing loss can collapse to a solution such that all but one high-level cause are ignored by the high-level mechanism (they have negligible influence on the target) and with that remaining cause corresponding to the one-cause solution (i.e. the solution you would get when training with only one high-level cause to begin with). This is arguably a valid solution to the TCR problem from the perspective of causal consistency, but not necessarily what we wish to achieve if we want to maximize the “explanatory” power of the reduction. The balancing regularization (10) penalizes such collapse by putting a preference on solutions that spread the causal influence on the target across multiple independent high-level causes. Furthermore, balancing, together with overlap regularization, provides a differentiable way to search for a good partition function $\pi$. This would be difficult to solve as a discrete optimization problem.

---

### Official Review · Reviewer_y9ou · 2024-03-21

**Q2-1 Originality-Novelty:** 3
**Q2-2 Correctness-Technical Quality:** 3
**Q2-5 Clarity Of Writing:** 4

**Q1 Summary And Contributions:**

The paper introduces a dimensionality-reduction technique for Linear Gaussian Structural Causal Models (SCMs) that is rooted in the Causal Abstraction literature. They propose an algorithm named Targeted Causal Reduction (TCR) that, assuming the existence of a target variable whose abstraction function is given, fits a user-defined number of abstract causes given interventional data. In other terms, by assuming to have a low-level model, even implicitly through a simulator, they propose to approximate an abstract model whose structure is known but the parameters are not.

**Q2-3 Extent To Which Claims Are Supported By Evidence:**

3: Good: the main claims are supported by convincing evidence (in the form of adequate experimental evaluation, proofs, (pseudo-)code, references, assumptions).

**Q2-4 Reproducibility:**

3: Good: key resources (e.g. proofs, code, data) are available and key details (e.g. proofs, experimental setup) are sufficiently well-described for competent researchers to confidently reproduce the main results.

**Q3 Main Strengths:**

The paper interestingly frames the difference between trying to learn causal abstractions and the more consolidated field of Causal Representation Learning. They then propose a strategy, named Targeted Causal Reduction (TCR), to fit an exact transformation from low-level interventional samples only and the high-level graph. While still requiring the abstract graph, their contribution is novel compared to existing abstraction learning approaches which either assumed abstract parameters or the explicit definition of the intervention map $\omega$. They then provide a sufficiently comprehensive experimental study on random and physics-inspired simulation data showing how, with a sufficiently large number of interventions, TCR recovers the exact transformation.

**Q4 Main Weakness:**

While initially motivating their solution through Causal Abstraction and consistently referring to the abstraction literature, the authors proceed to define the relation between their models as an Exact Transformation (Rubenstein et al., 2017). Since all abstractions are exact transformations, as detailed by Beckers et al. (2009), it is not incorrect to state that they are approximating such transformations. However, the paper misses a connection with the theory of Causal Abstraction with Soft Interventions (Massidda et al., 2023), which would have effectively characterized the learned relation as a causal abstraction. (*This point has been addressed during the rebuttal.*)

**Q5 Detailed Comments To The Authors:**

- In Definition 2.1, what do the authors intend with "$\mathbf{g}$ $P_{\mathbf{U}}$"? I would have expected $\mathbf{g}$ to be measurable, since by being a prob. density $P_{\mathbf{U}}$ should always be measurable.
- In Proposition 3.4 the shape of ${\bar{\omega_1}}$ is not well defined, since it is the multiplication of an object of size $|\Omega| \times |\Omega|$ with a vector $N_1$ . I guess it should be $(I_{11} - A_{11})$ , as it is reported in Equation 16 from the Proof in Appendix.
- In the introduction, the paper references (Geiger and Straehle, 2020). I admittedly never read the paper and just skimmed it through after seeing your reference, but I can't see the connection with the sentence. Is it a citation error by any chance?
- In Algorithm 1, what does the LCPR acronym stand for?
- In the paper, you characterize your identifiability results for exact transformations. Do you have any proof or counterexample to show that the relation is indeed not a causal abstraction in the case of Linear Gaussian models?

**Q9 Complying With Reviewing Instructions:**

Yes

---

> ### Author Rebuttal · Authors · 2024-04-04
>
> We are thankful for the in-depth review and the dedication of your time towards our work. It is a great opportunity for us to make necessary clarifications and to enhance our paper.
>
> ### **Reply to Q4**
>
> Thank you for bringing up the relationship to causal abstractions of our theory section. We have used your suggestion to provide an additional result in the paper. We discuss this in more detail below.
>
> ### **Answers to the detailed comments**
>
> *Assumption on $\boldsymbol{g}$ In Definition 2.1*
>
> Here, we mean measurability of $\boldsymbol{g}$ relative to the sigma algebra of $P_U$ and you are correct. The statement is not necessary. We will simply put “$g$ measurable” in the camera-ready version.
>
> *“In Proposition 3.4 …”*
>
> The notation could be a bit clearer here. We apologize for that. For the 1-cause case, the whole set of all intervened variables is used to compute the single cause, such that $pi(1)=\Omega$ and $A_{\Omega, \Omega}$ refers to the submatrix of the adjacency of size $N_1$ by $N_1$ and $I_\Omega$ is the identity matrix of size $N_1$. So the dimensions match the vector it is multiplied with. In the proof in the appendix we use $A_{11}$ for $A_{\Omega, \Omega}$, since $\pi(1)=\Omega$ in the 1-cause case. We will clarify this statement in the camera-ready version.
>
> *Geiger et al. citation*
>
> Thank you for pointing this out. The citation should be the reference labeled Geiger 2023a.  This will be corrected in the camera-ready version.
>
> *Meaning of LCPR*
>
> This was a typo, it should be LTCR. This will be corrected.
>
> *Link to causal abstractions*
>
> We thank the reviewer for pointing out the connection with soft causal abstractions, as studied in the work of Massidda et al. (2023) currently cited in our appendix, that can be made for the theoretical result of Proposition 3.4. To provide some background for other reviewers, causal abstractions were introduced by Beckers (2019) as a special case of exact transformations satisfying additional constraints. In short, instead of matching low-level with high-level interventional distributions, an abstraction also requires matching the interventional maps from exogenous to endogenous variables at both levels. Our paper focuses on learning approximately exact transformation, by minimizing the mismatch between the relevant interventional distribution. For this purpose, the theory part exhibits a setting where there exists a unique exact transformation (Proposition 3.4), which can be learned using our algorithm, and allows us to validate the approach using synthetic experiments. However, it turns out that the unique solution provided in Proposition 3.4 can also be turned into a causal abstraction, as exhibited in the Proposition D.2 available at the following [link](https://figshare.com/s/f5bd146e4a6700bf1c28). This result is interesting for researchers investigating the connection between the two frameworks, so we suggest adding the following main text sentence after Proposition 3.4.
> “The resulting constructive transformation can also be associated to a constructive causal abstraction under the same model assumptions, as shown in Proposition D.2. In particular, the absence of ambiguity of the unique interventional map exhibited in our result is consistent with the condition formulated by Massidda et al. (2023) for soft abstractions: if soft interventions performed at the low level map to hard interventions at the high level, then the intervention map omega can be unambiguously defined. This is the case for our setting because shift interventions at the high-level are performed only on root nodes, making them valid hard interventions.”
>
> This also suggests that linear models with shift interventions are an interesting special setting where exact transformations and causal abstractions may coincide. The remainder of our framework is, however, designed to learn an approximation of exact transformations, while learning (approximate) causal abstractions would require a different approach left to future work.

---

### Official Review · Reviewer_SBy5 · 2024-03-22

**Q2-1 Originality-Novelty:** 3
**Q2-2 Correctness-Technical Quality:** 3
**Q2-5 Clarity Of Writing:** 3

**Q1 Summary And Contributions:**

The papaer proposes a method to condense complex interventional models into a concise set of causal factors that explain a specifc target phenomenon by using an information theoretic objective from interventions of simulation. It proposes an algorithm to learn TCRs and establish identifiability for continuous variables under shift interventions. The paper argues that the existing causal model reduction methods use discrete variables and rely on on hard interventions in general.

**Q2-3 Extent To Which Claims Are Supported By Evidence:**

3: Good: the main claims are supported by convincing evidence (in the form of adequate experimental evaluation, proofs, (pseudo-)code, references, assumptions).

**Q2-4 Reproducibility:**

3: Good: key resources (e.g. proofs, code, data) are available and key details (e.g. proofs, experimental setup) are sufficiently well-described for competent researchers to confidently reproduce the main results.

**Q3 Main Strengths:**

- It gives a motivating example why there is a need for a much simpler high-level causal models for continuous variables and soft interventions in some domains.
- The paper is fairly well-written.
- The paper has made good contributions to an unexplored area in causal model reduction.

**Q4 Main Weakness:**

- The proposed method is limited to linear reduction that is built on strong assumptions.
- Assumption 3.3 bypasses one of the main challenges in the entire idea of causal model reduction.

**Q5 Detailed Comments To The Authors:**

1. What is the difference between the reduced low-level interventional distribution and the fitted high-level interventional model? I thought the low-level interventional distribution is reduced to a high-level interventional model?
2. What do $l, n$ represent in definition 2.3?
3. Typo: focussing, equation 5 seems to have double closing prostheses..
4. Why should one expect to not have confounding effects and cycles in the high-level model when they are allowed in the low-level model?
5. In the paper, it says  “Assuming non-cancellation of causal paths is akin to preventing faithfulness violations and generically satisfied” What is the theoretical basis to say that non-cancellation is generically satisfied?

**Q9 Complying With Reviewing Instructions:**

Yes

---

> ### Author Rebuttal · Authors · 2024-04-04
>
> Thank you for the feedback and the time you have invested in reviewing our paper. We appreciate the opportunity to clarify and enhance our work based on your observations.
>
> ### **Reply to Q4**
>
> The goal of our approach is to learn high-level explanations of a complex low-level simulation. We argue that simplifying assumptions are necessary for technical reasons (for the Gaussian assumption) and, more importantly, to obtain interpretable reductions. We give a more detailed answer in the response to Reviewer 2hcr.
>
> Assumption 3.3 gives us insights into the minimal number of distinct interventions needed to find a solution with perfect consistency for a linear low-level SCM. We agree that it would be desirable to have guarantees for fewer interventions, but how to achieve it remains an open question. However, we also think that this assumption is not prohibitive for the simulation setting that we are interested in. Typically, one has access to the data generating process and can sample under new interventions. Then, it becomes a question of compute resources as the number of low-level variables grows. We hope to investigate how to extend our framework to allow a more limited set of interventions in future work.
>
> ### **Answers to the detailed comments**
>
> 1. *“Difference between the reduced low-level interventional distribution [..] and the fitted high-level”*
>
> If the learned reduction is an exact transformation (Definition 2.2), then there is no difference between these two distributions. The discrepancy between them is our main learning signal through the consistency loss of equation (4). Some settings (Proposition 3.4) provide guaranties for the distributions to match exactly once the model is learned; otherwise, a mismatch may remain after learning.
>
> 2. *“What do $l$ and $n$ represent in definition 2.3?”*
>
> Thank you for pointing this out. There is a typo in the definition: it should be $l$ instead of $n$. $k$ and $l$ represent index sets corresponding to subsets of the low-level input. The statement with $k$ and $l$ says that for constructive transformations, these subsets should be non-overlapping. In other words, each of the high-level variables attend to a different subset of the low-level variables. We made this choice to ensure that the high-level variables correspond to distinct causes of the target and there is no mixing between them, helping interpretability of the learned representation.
>
> 3.  *“Typo: focussing, equation 5 seems to have double closing parentheses”*
>
> Thank you for pointing out the typo. It will be corrected in the camera-ready version. Equation 5 runs over two lines, so the last closing bracket belongs to the first expectation in the line above. We will try to make this visually clearer.
>
> 4.  *“Why should one expect to not have confounding effects and cycles in the high-level model when they are allowed in the low-level model?”*
>
> The high-level causal model with independent causes and the linear $\tau$- and $\omega$-maps between variables form a restricted model class with the advantage that the found reductions are straightforward to interpret. We fit this model such that interventions on the low-level are as consistent as possible with interventions in the high-level model. We show that under some additional restrictions we can achieve perfect consistency and guarantee uniqueness of the solution. But, in general, the fitted model is only a simpler approximation to a more complex ground truth. In principle, fitting a more complex high-level model could be done, but it would make it more difficult to interpret, since you would have to consider more complex interdependencies of variables. Furthermore, how to guarantee that these more complex models would lead to unique reductions is an interesting open question, beyond the scope of the present work.
>
> 5. *“What is the theoretical basis to say that non-cancellation is generically satisfied?”*
>
> This relates to the argument of genericity of faithfulness in linear models proposed by Spirtes et al. [*Causation, Prediction and Search*, 2000, Theorem 3.2]. In short: the condition for having path cancellation takes the form of a polynomial equation in the causal coefficients, such that the associated subset of model parameters is an algebraic variety with zero Lebesgue measure. Under the assumption of sampling model coefficients from a joint distribution with a density with respect to the Lebesgue measure, models with path cancellation happen with probability zero. We will provide this citation and argument in the final version.

---

### Official Review · Reviewer_2hcr · 2024-03-24

**Q2-1 Originality-Novelty:** 3
**Q2-2 Correctness-Technical Quality:** 3
**Q2-5 Clarity Of Writing:** 3

**Q1 Summary And Contributions:**

Understanding why phenomena occur is fundamental to scientific inquiry, often relying on simulations of scientific models. As models become more complex, unraveling the causes behind phenomena in high-dimensional spaces of interconnected variables becomes increasingly difficult. This paper propose Targeted Causal Reduction (TCR) as a method to condense complex intervenable models into a concise set of causal factors explaining a specific target phenomenon. It introduces an information-theoretic objective for learning TCR from interventional data of simulations, establish identifiability for continuous variables under shift interventions, and present a practical algorithm for learning TCRs. It demonstrate its ability to generate interpretable high-level explanations from complex models on toy and mechanical systems, illustrating its potential to aid scientists in studying complex phenomena across various disciplines.

**Q2-3 Extent To Which Claims Are Supported By Evidence:**

1: Poor: the authors fail to convincingly backup their main claims (e.g., if the experimental evaluation is flawed, proofs are lacking or invalid, references are missing, assumptions are not realistic, not specified, or not motivated).

**Q2-4 Reproducibility:**

1: Poor: key details (e.g. proof sketches, experimental setup) are incomplete/unclear, or key resources (e.g. proofs, code, data) are unavailable.

**Q3 Main Strengths:**

This paper proposes a motivating framework for reducing the inherent complexity of a causal model to a high-level expression that retains the qualities needed to answer causal queries.  The work seems solid and mathematically sound.

**Q4 Main Weakness:**

It is unrealistic to expect a conference reviewer to spend the time required to fully understand these 10 dense pages of paper and 16 dense pages of appendices.  This work deserves to be analyzed as a journal article.

Similarly, it seems to me that publishing the 10-pages article without the appendices doesn't really make sense. Whether it's the state of the art, or some of the demonstrations (for example, that of proposition 3.1), they deserve to be read as much as the rest of the article.

The algorithm seems so abstract that there's not much of interest left (spending 4 lines incrementing lists ...).
The difference between the '<-' operator and the '=' operator isn't clear either.

Given the density of the paper, it's a little difficult to highlight any gaps. Still, the assumptions (linearity, normality, shift intervention, etc.) seem possibly restrictive and a discussion (or even just a justification) on that matter would certainly be interesting (more than just stated in the "Limitations").

**Q5 Detailed Comments To The Authors:**

In conclusion, this paper seems to me too interesting not to give it a good score, but on the other hand, it seems to me that it won't be of much interest once the future full-length journal article will be published.

**Q9 Complying With Reviewing Instructions:**

Yes

---

> ### Author Rebuttal · Authors · 2024-04-04
>
> We thank you for taking the time to review our paper and provide feedback.
>
> ### **Answer to Q4**
>
> *On Algorithm 1*
>
> Thank you for pointing out the inconsistency in the used operators. There is a typo in the last line of the most inner for-loop: it should be assigned (using <-). We will also shorten the assignments there into one line for the three lists. We agree that the shown algorithm is rather abstract. Our main goal here was to show how sampling interventions and simulations relate to the training loop. For example, for a given intervention, $B$ samples of the simulation are generated.
>
> *On the assumptions*
>
> The assumption of Gaussian distributions has a technical reason: we make this approximation in order to make the KL divergence in the consistency loss (5) tractable. Admittedly, it is a strong assumption, but experimentally we have observed that TCR finds reasonable reductions even when the involved distributions deviate from perfect Gaussians. In the double well experiment, the high-level cause distribution with and without intervention is shown in Figure 4b. Generalizing the framework to nonparametric distributions is left to future work.
>
> Another technical design choice is the shift interventions. The use of shift interventions lends itself to theoretical analysis (Prop. 3.4) and allows us to intervene on multiple variables simultaneously such that  their effect propagate to the target without having to know the details about the causal structure of the low-level model. We give more details on this point in the response to Reviewer Ufeg.
>
> The other assumptions serve our main goal of interpretability. Simplification is key to understanding complex simulations, and to this end, we constrain our model with linear $\tau$-/$\omega$-maps and assume independent high-level causes. The weights from linear maps serve as indicators of influential aspects of the input on the target phenomenon. For interpreting higher capacity nonlinear maps, we would have to rely on additional post-hoc interpretability tools. Our independence assumption is key to disentangling distinct ground truth causes when those exist, giving an accessible, reduced representation similar to the principles of Principal Component Analysis (PCA). Our goal here is not necessarily to precisely replicate the simulation. For this, we already have access to the simulation itself.

---

### Meta-Review · Area_Chair_PRG2 · 2024-04-21

The paper proposes a motivating framework for reducing the inherent complexity of a causal model to a high-level expression that retains the qualities needed to answer causal queries. All reviewers found the work solid and mathematically sound.